# HASSOD: Hierarchical Adaptive Self-Supervised Object Detection

**Shengcao Cao[1]   Dhiraj Joshi[2]   Liang-Yan Gui[1]   Yu-Xiong Wang[1]**
[1]University of Illinois at Urbana-Champaign   [2]IBM Research
[1]{cao44,lgui,yxw}@illinois.edu   [2]djoshi@us.ibm.com

## Abstract

The human visual perception system demonstrates exceptional capabilities in learning without explicit supervision and understanding the part-to-whole composition of objects. Drawing inspiration from these two abilities, we propose Hierarchical Adaptive Self-Supervised Object Detection (HASSOD), a novel approach that learns to detect objects and understand their compositions without human supervision. HASSOD employs a hierarchical adaptive clustering strategy to group regions into object masks based on self-supervised visual representations, adaptively determining the number of objects per image. Furthermore, HASSOD identifies the hierarchical levels of objects in terms of composition, by analyzing coverage relations between masks and constructing tree structures. This additional self-supervised learning task leads to improved detection performance and enhanced interpretability. Lastly, we abandon the inefficient multi-round self-training process utilized in prior methods and instead adapt the Mean Teacher framework from semi-supervised learning, which leads to a smoother and more efficient training process. Through extensive experiments on prevalent image datasets, we demonstrate the superiority of HASSOD over existing methods, thereby advancing the state of the art in self-supervised object detection. Notably, we improve Mask AR from 20.2 to **22.5** on LVIS, and from 17.0 to **26.0** on SA-1B. Project page: https://HASSOD-NeurIPS23.github.io.

## 1   Introduction

The development of human visual perception is remarkable for two key abilities: 1) Humans begin learning to perceive objects in their environment through observation alone [25], without needing to learn the names of these objects from external supervision. 2) Moreover, human perception operates in a hierarchical manner, enabling individuals to recognize the part-to-whole composition of objects [2, 23]. These characteristic capabilities offer valuable insights into the learning processes of object detectors, which still heavily rely on the availability and quality of fine-grained training data. For example, the state-of-the-art detection/segmentation model, Segment Anything Model (SAM) [18], is developed on a dataset of 11 million images and 1 billion object masks. It remains an open question how to effectively learn to detect objects and recognize their compositions from even larger-scale datasets (*e.g.*, LAION-5B [26]) without such object-level annotations.

In prior work on self-supervised object detection [37, 38], a two-stage *discover-and-learn* paradigm is adopted: 1) Self-supervised visual representations [5, 15] are obtained, and a saliency-based method is employed to extract the most prominent one or few objects. 2) Subsequently, an object detector is trained based on these pseudo-labels, sometimes involving multiple rounds of self-training for refinement. However, despite such attempts to eliminate the need for external supervision, several weaknesses persist in these approaches: 1) **Narrow coverage of objects.** The focus on only one or few objects per image in previous methods undermines their ability to fully exploit the learning signals in

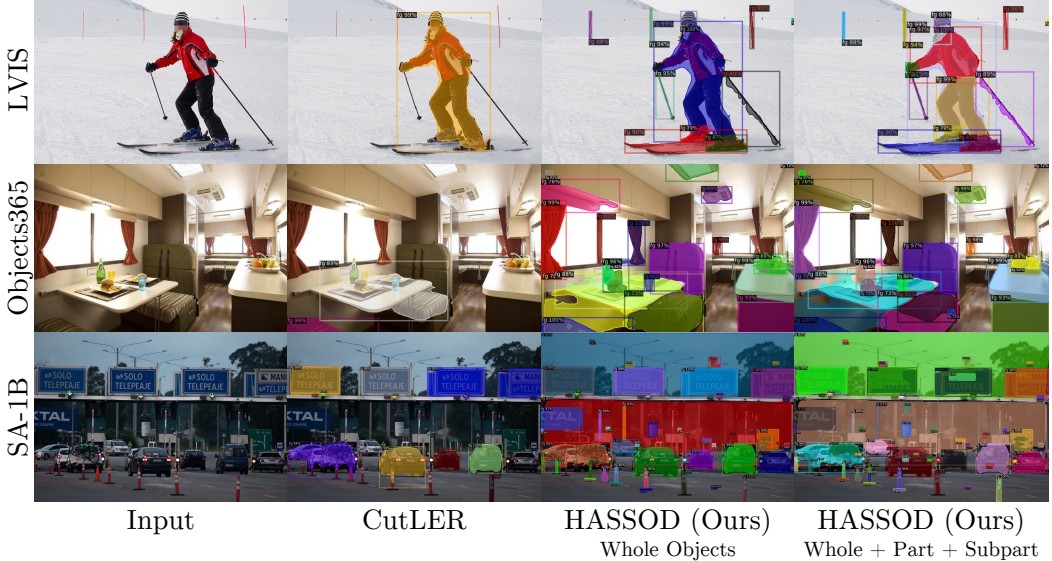

| Input | CutLER | HASSOD (Ours)
Whole Objects | HASSOD (Ours)
Whole + Part + Subpart |

Figure 1: Fully self-supervised object detection and instance segmentation on prevalent image datasets. Our approach, HASSOD, demonstrates a significant improvement over the previous state-of-the-art method, CutLER [38], by discovering a more comprehensive range of objects. Moreover, HASSOD understands the part-to-whole object composition like humans do, while previous methods cannot.

natural scene images containing dozens of objects, such as those in the MS-COCO dataset [20]. This narrow focus also restricts the capability of these methods to accurately detect and segment multiple objects within an image. 2) **Lack of composition.** Prior work often overlooks the composition of objects, neglecting the identification of hierarchical levels for whole objects, part objects, and subpart objects (*e.g.*, considering an image of a bicycle; the bicycle is a whole object, its wheels and handles are parts, and the spokes and tires are subparts). This oversight not only limits the interpretability of learned object detectors, but also hinders the model's ability to tackle the intrinsic ambiguity in the task of segmentation. 3) **Inefficiency.** The reliance on multi-round self-training in earlier methods can result in inefficient and non-smooth training processes, which further constrains the potential of self-supervised object detection and comprehension of object composition.

Inspired by the unsupervised, hierarchical human visual perception system, we propose Hierarchical Adaptive Self-Supervised Object Detection (HASSOD), *aiming to address these limitations* mentioned above and better harness the potential of self-supervised object detection, as depicted in Figure 1. First, unlike previous methods that limit their focus to one or few prominent objects per image, HASSOD employs a hierarchical adaptive clustering strategy to group regions into object masks, based on self-supervised visual representations. By adjusting the threshold for terminating the clustering process, HASSOD is capable of effectively determining the appropriate number of objects per image, thus better leveraging the learning signals in images with multiple objects.

The second key component of HASSOD is its ability to identify hierarchical levels of objects in terms of composition. By analyzing the coverage relations between masks and building tree structures, our approach successfully classifies objects as *whole* objects, *part* objects, or *subpart* objects. This novel self-supervised learning task not only improves detection performance, but also enhances the interpretability and controllability of the learned object detector, a property that prior self-supervised detectors lack. Therefore, HASSOD users can comprehend how the detected whole objects are assembled from smaller constituent parts. Simultaneously, they can control HASSOD to perform detection at their preferred hierarchical level, thereby catering more effectively to their needs.

Finally, HASSOD abandons the multi-round self-training used in previous methods which lacks efficiency and smoothness. Instead, we take inspiration from the Mean Teacher [22, 31] framework in semi-supervised learning, employing a teacher model and a student model that mutually learn from each other. This innovative adaptation facilitates a smoother and more efficient training process, resulting in a more effective self-supervised object detection approach.

In summary, the key contributions of HASSOD include:

- A hierarchical adaptive clustering strategy that groups regions into object masks based on self-supervised visual representations, adaptively determining the number of objects per image and effectively discover more objects from natural scenes.
- The ability to identify hierarchical levels of objects in terms of composition (whole/part/subpart) by analyzing coverage relations between masks and building tree structures, leading to improved detection performance and enhanced interpretability.
- A novel adaptation of the Mean Teacher framework from semi-supervised learning, which replaces the multi-round self-training in prior methods, leading to smoother and more efficient training.
- State-of-the-art performance in self-supervised object detection, enhancing Mask AR from 20.2 to **22.5** on LVIS [11], and from 17.0 to **26.0** on SA-1B [18]. Remarkably, these results are achieved through training with only $1/5$ of the images and $1/12$ of the iterations required by prior work.

## 2 Related Work

**Unsupervised object detection/discovery.** Identifying and locating objects in images without using any human annotations is a challenging task, as it requires learning the concept of objects from image data without any external supervision. OSD [33] formulates this task as an optimization problem on a graph, where the nodes are object proposals generated by selective search, and the edges are constructed based on visual similarities. rOSD [34] improves the scalability of OSD with a saliency-based region proposal algorithm and a two-stage strategy. LOD [35] formulates unsupervised object discovery as a ranking optimization problem for improved computation efficiency. Following the observation that DINO [5], a self-supervised pre-training method, can segment the most prominent object in each image, LOST [29], FOUND [30], and FreeSOLO [37] train object detectors using saliency-based pseudo-labels. TokenCut [39] and CutLER [38] also use self-supervised representations, but generate pseudo-labels by extending Normalized Cuts [28]. Saliency-based region proposal and Normalized Cuts are both focused on the prominent objects in each image, and usually only propose one or few objects per image. Different from these approaches, HASSOD produces initial pseudo-labels using a hierarchical adaptive clustering strategy, which can adaptively determine the number of objects depending on the image contents.

**Object detection by parts.** Detecting objects by identifying their composing parts has been widely studied in computer vision. Deformable Parts Model (DPM) [9] is a seminal approach that utilizes discriminatively trained part-based models for object detection, which effectively models complex object structures and improves over monolithic detectors. A following method [6] not only detects the objects, but also simultaneously represents them using body parts, highlighting the importance of both holistic models and part-based representations. This idea is extended by leveraging both whole object and part detections to infer human actions and attributes [10], suggesting the advantage of a combined approach. In this work, we revisit this classic idea of representing and detecting whole objects as well as their parts in the context of self-supervised learning.

## 3 Approach

In this section, we introduce the learning process in our proposed approach, Hierarchical Adaptive Self-Supervised Object Detection (HASSOD). Following prior work on unsupervised object detection [29, 30, 37–39], HASSOD adopts a two-stage *discover-and-learn* process to learn a self-supervised object detector, as illustrated in Figure 2. In the first stage, we discover objects from unlabeled images using self-supervised representations, and generate a set of initial pseudo-labels. Then in the second stage, we learn an object detector based on the initial pseudo-labels, and smoothly refine the model by self-training. The first stage is based on pre-trained, fixed visual features, and the second stage learns an object detector to improve over the fixed visual features and pseudo-labels. In the following subsections, we describe the three core components of HASSOD in detail.

### 3.1 Hierarchical Adaptive Clustering

In the first stage, HASSOD creates a set of pseudo-labels as the initial self-supervision source. We propose a hierarchical adaptive clustering strategy to discover object masks as pseudo-labels, using only unlabeled images and a frozen self-supervised visual backbone. Figure 3 provides an overview of this procedure. Our hierarchical adaptive clustering algorithm extends agglomerative

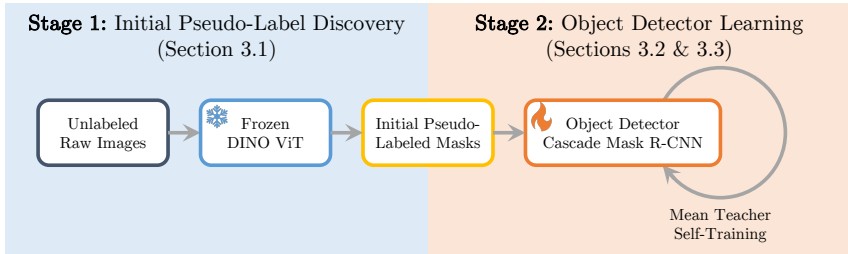

Figure 2: Two-stage discover-and-learn process in HASSOD. Stage 1 uses a frozen, self-supervised DINO [5] ViT backbone to discover initial pseudo-labels from unlabeled images. Stage 2 learns an object detector to improve over the pre-trained features and initial pseudo-labels.

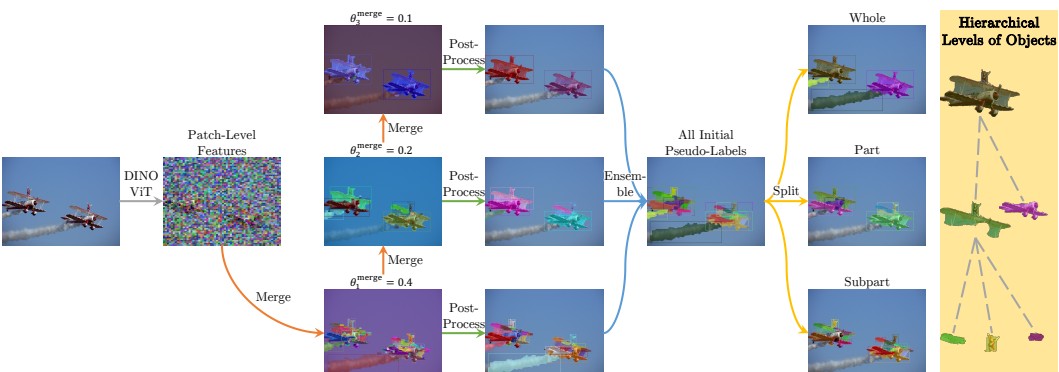

Figure 3: Hierarchical adaptive clustering and hierarchical levels of objects. The procedure of creating initial pseudo-labels for training the object detector without any human annotations includes the following steps: (Initialize) Visual features are extracted from the given image by a ViT pre-trained with DINO [5], and each $8 \times 8$ patch is initialized as one individual region. (Merge) Adjacent regions with the highest feature similarities are progressively merged into object masks, until the pre-set thresholds $\theta_i^{\text{merge}}$ are reached. (Post-Process) Object masks are selected and refined using simple post-processing techniques. (Ensemble) Results from multiple thresholds $\{\theta_i^{\text{merge}}\}_{i=1}^3$ are combined to ensure better coverage of potential objects. (Split) Analysis of coverage relations divides objects into three hierarchical levels: whole, part, and subpart. The example on the right illustrates the tree structure of object composition: The whole aircraft is composed of an upper and a lower part. The upper part further consists of a left wing, a right wing, and a person standing on it.

clustering [12], grouping adjacent image patches into semantically coherent masks based on the similarity of self-supervised visual representations. More specifically, we use a *frozen* ViT-B/8 model [8] pre-trained on unlabeled ImageNet [7] by DINO [5], a self-supervised representation learning method, to extract visual features. For each image, we take the feature map generated by this model at its final Transformer layer [32]. Each spatial element in the feature map corresponds to a $8 \times 8$ patch in the original image.

To initiate the hierarchical adaptive clustering process, we treat each patch as an individual region. We then compute the pairwise cosine similarity between the features of adjacent regions to measure their closeness in the semantic feature space. The regions are gradually merged into masks that represent objects by iteratively performing the following steps: 1) Identify the pair of adjacent regions with the highest feature similarity. 2) If the similarity is smaller than the pre-set threshold $\theta^{\text{merge}}$, stop the merging process. 3) Merge the two regions, and compute the feature of the merged region by averaging all the patch-level features within it. 4) Update the pairwise similarity between this newly merged region and its neighbors. This merging process is visualized in Figure 3, columns 1-3.

Once the merging process is complete, we perform a series of automated post-processing steps to refine and select the masks, including Conditional Random Field (CRF) [19] and filtering out masks that are smaller than 100 pixels or contain more than two corners of the image. These steps are based on standard practices in previous work [38] and require no manual intervention. Our hierarchical adaptive clustering strategy effectively groups regions into object masks based on self-supervised visual representations, adaptively determining the appropriate number of objects per image. In

images containing multiple objects with heterogeneous semantic features, the merging process stops earlier, resulting in a larger number of regions corresponding to different objects. Conversely, in highly homogeneous images, more regions are merged, leading to fewer object masks. This adaptive approach enables HASSOD to cover more objects for self-supervised learning, rather than being limited by one or a few prominent objects in each image in prior work [38, 39].

In practice, we are not restricted to one single fixed threshold $\theta^{\mathrm{merge}}$ to determine the stopping criterion for the clustering process. Instead, we find it beneficial to ensemble results from multiple (*e.g.*, 3) pre-set thresholds $\{\theta_i^{\mathrm{merge}}\}_{i=1}^3$. When the currently highest feature similarity reaches one of these thresholds, we record the derived object masks from the merged regions at that step. Utilizing multiple thresholds allows us to capture objects of various sizes and at different hierarchical levels of composition, enabling a more comprehensive coverage of objects in scene images. The post-processing and ensemble are visualized in Figure 3, columns 4-5.

## 3.2 Hierarchical Level Prediction

In the following second stage, HASSOD learns an object detection and instance segmentation model, *e.g.*, Cascade Mask R-CNN [4], using the initial pseudo-labels generated in the first stage. By training on such pseudo-labels, the model learns to recognize common objects across different training images, and thus achieves *enhanced generalization* to images which the model has not seen during training.

In addition to the standard object detection objective, we aim to equip our detector with the ability to understand the hierarchical structure among objects and their constituent parts. In HASSOD, we incorporate the concept of *hierarchical levels* into object masks by leveraging the *coverage relations* between them. Formally, we say mask A is covered by mask B when three conditions are satisfied (with respect to a pre-set coverage threshold $\theta^{\mathrm{cover}}\%$): 1) More than $\theta^{\mathrm{cover}}\%$ of pixels in mask A are also in mask B. 2) Less than $\theta^{\mathrm{cover}}\%$ of pixels in mask B are in mask A. 3) Mask B is the smallest among all masks satisfying the previous two conditions. Intuitively, if mask B covers mask A, it suggests that A is a part of B and B is at a higher level than A. If we consider A and B as tree nodes, A should be a child of B. Using all such coverage relations, we can construct a *forest of trees* that contain all masks in an image. Ultimately, the roots of all trees in this image are considered as "whole" objects, their direct children are "part" objects, and all the remaining descendants are "subpart" objects. An example is shown on the right side of Figure 3.

After identifying the hierarchical levels of object masks in the pseudo-labels, we attach a new classification head to the object detector for level prediction, which classifies each predicted object as a whole object, a part object, or a subpart object. This new component enables HASSOD to model object composition effectively, resulting in improved object detection performance and enhanced interpretability compared with previous self-supervised object detection methods. The hierarchical level prediction head is added alongside the existing foreground/background classification head, box regression head, and mask prediction head. Subsequently, we train the object detector using the initial set of object mask pseudo-labels obtained from the hierarchical adaptive clustering process, as well as the additional level prediction task.

## 3.3 Mean Teacher Training with Adaptive Targets

Notably, the initial pseudo-labels derived in the first stage contain noise and are not perfectly aligned with real objects. To improve over such noisy pseudo-labels, prior work [37, 38] usually employs multi-round self-training to refine the model, *i.e.*, using a well-trained detector to re-generate pseudo-labels and re-train a new detector. *For the first time*, HASSOD refines the object detector efficiently and smoothly by adapting the Mean Teacher learning paradigm [22, 31] from semi-supervised learning to the fully self-supervised setting.

Before introducing our innovative adaptation of Mean Teacher in the self-supervised setting, we first briefly summarize the mutual-learning process in Mean Teacher (see Figure 4). Mean Teacher employs two models, a teacher and a student, which learn from each other. The teacher takes weakly-augmented, unlabeled images as input and provides detection outputs as training targets for the student. The student's weights are updated to minimize the discrepancy between its predictions and the targets given by the teacher on the same unlabeled images but with strong augmentation. In the semi-supervised setting, the student receives supervision from two sources simultaneously. One source is the "teacher-to-student" branch mentioned above, and the other is the "label-to-student" branch

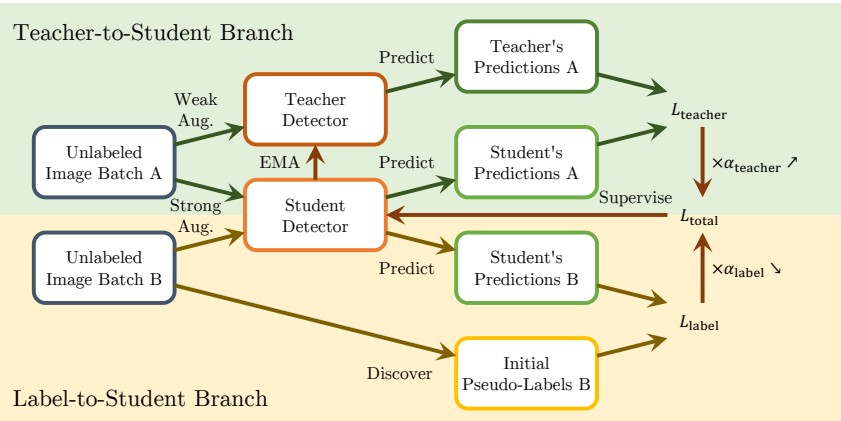

Figure 4: Mean Teacher self-training with adaptive targets in HASSOD. Two detectors of the same architecture, the teacher and the student, learn from each other to improve over the initial pseudo-labels. The teacher is updated as the exponential moving average (EMA) of the student. The student receives supervision from two branches: The teacher-to-student branch (**top**) encourages the student to mimic the teacher's predictions; the label-to-student branch (**bottom**) minimizes the discrepancy between the student's predictions and the initial pseudo-labels. During training, our proposed adaptive target strategy increases the weight for the teacher-to-student branch, $\alpha_{\text{teacher}}$, and decreases the weight for the label-to-student branch, $\alpha_{\text{label}}$, since the teacher becomes a more and more reliable self-supervision source compared with the initial pseudo-labels.

where the student learns from images with ground-truth labels. Both branches compute standard detection losses (*e.g.*, bounding box classification and regression), and the student is optimized to minimize the total loss. The teacher's weights are an exponential moving average of the student's weights, ensuring smooth and stable training targets.

In HASSOD, we do not have any labeled images from human supervision but instead utilize two sources of self-supervision. One source is the initial pseudo-labels obtained from our hierarchical adaptive clustering, which functions similarly to the labels-to-student branch in the semi-supervised setting. The other source is the detection predictions made by the teacher model, which corresponds to the teacher-to-student branch in Mean Teacher. Different from standard Mean Teacher, our method employs adaptive training targets, as we gradually adjust the loss weights for the two branches. This is because the initial pseudo-labels may not effectively cover all possible objects, while the teacher model will progressively improve as a better source of supervision. Consequently, during Mean Teacher self-training, we continuously decrease the loss weight $\alpha_{\text{label}}$ for the branch that uses the initial pseudo-labels and increase the loss weight $\alpha_{\text{teacher}}$ for the branch based on the teacher's predictions, following a cosine schedule.

## 4 Experiments

In this section, we conduct extensive experiments to evaluate HASSOD in comparison with previous methods. We first describe the training details and efficiency in Section 4.1. Section 4.2 introduces the datasets and metrics used for evaluation. Section 4.3 presents our main results of self-supervised object detection and instance segmentation. Section 4.4 provides some qualitative results and analysis. Section 4.5 conducts further experiments to verify the effects of each component in HASSOD. Additional quantitative and qualitative results are included in the appendix.

### 4.1 Data-Efficient and Computation-Efficient Training

We train a Cascade Mask R-CNN [4] with a ResNet-50 [13] backbone on MS-COCO [20] images. The backbone is initialized from DINO [5] self-supervised pre-training. We use both the `train` and `unlabeled` splits of MS-COCO, totaling to about 0.24 million images. Notably, this amount of images is only **1/5** of ImageNet used by prior work CutLER [38]. Compared with ImageNet-like [7] iconic images, images in MS-COCO are mostly captured in complex scenes containing multiple objects with diverse layouts and compositions. Therefore, each image offers richer learning resources for object detectors, *enabling effective detector training with significantly fewer images*. The whole

training process spans 40,000 iterations, taking about 20 hours on 4 NVIDIA A100 GPUs. The efficiency and smoothness introduced by the Mean Teacher self-training approach reduces the training iterations to $1/12$ of that required by CutLER [38], highlighting the computation efficiency of our training strategy. Implementation details are included in Appendix J.

## 4.2 Evaluation Datasets and Metrics

We mainly conduct our experiments in a *zero-shot* manner on the validation sets of three benchmark datasets, namely Objects365 [27], LVIS [11], and SA-1B [18]. Given that self-supervised object detection methods, including HASSOD, do not utilize class labels as a form of supervision, we follow prior work [37–39] and evaluate these models as *class-agnostic* detectors, comparing them only against the bounding boxes and masks provided in the dataset annotations.

- Objects365 [27] is a large-scale object detection dataset containing 365 object categories. The combined validation sets of Objects365 v1 and v2 include 80,000 images in total.
- LVIS [11] is a dataset that features a wide variety of over 1,200 object classes, using the same images as MS-COCO [20]. LVIS v1.0 validation set has 19,809 images, each annotated with object masks for instance segmentation.
- SA-1B [18] is a recent dataset that includes 11 million images and 1 billion fine-grained, model-generated object masks. SA-1B provides a more comprehensive coverage of all potential objects, facilitating a more robust evaluation of self-supervised object detectors. As SA-1B does not provide a validation split, we utilize a random subset of 50,000 images for our assessment.

In terms of evaluation metrics, we focus primarily on average recall (AR) rather than average precision (AP). The choice of AR over AP is motivated by the nature of the self-supervised task. In a dataset with a fixed number of classes, objects not labeled by humans – simply because they do not fall under the designated classes – may still be detected by a self-supervised detection model. Standard AP calculation would penalize such predictions as false positives, despite them being valid detections. In contrast, AR does not suffer from this issue, making it a more appropriate metric for our context. By prioritizing recall, we can more accurately assess the ability of our model to identify all relevant objects in an image, which aligns with the goal of the self-supervised object detection task. We evaluate AR based on both bounding boxes for object detection ("Box AR") and masks for instance segmentation ("Mask AR"). Appendix A discusses the evaluation metrics in detail.

## 4.3 Self-Supervised Detection and Segmentation

After we use HASSOD to train the object detection and instance segmentation model, Cascade Mask R-CNN, on MS-COCO images, we evaluate our model on Objects365, LVIS, and SA-1B datasets in a zero-shot manner, *i.e.*, no further training on these three datasets. The whole training and evaluation process is repeated for three times, and we report the mean performance for conciseness. The *standard deviation of AR is less than* 0.6 on all three datasets. Complete evaluation results are included in Appendix H.

We compare HASSOD with prior state-of-the-art self-supervised object detection methods, including FreeSOLO [37] and CutLER [38]. We also include results from SAM [18], the latest *supervised* class-agnostic detection/segmentation model, to gain understanding of the gap between self-supervised and supervised models, and how HASSOD is effectively closing this gap. To be consistent with other models, we provide SAM with only the raw images but no bounding boxes or points as prompts. For prior methods FreeSOLO, CutLER, and SAM, we directly evaluate the publicly available model checkpoints on the given datasets. Considering that the number of ground-truth labels per image may be greater than 100, we allow all models to output up to 1,000 predictions per image.

As shown in the main results summarized in Table 1, HASSOD significantly improves the detection and segmentation performance over previous self-supervised models FreeSOLO and CutLER. On Objects365, we improve the Box AR by 3.2; on LVIS, we improve Box AR by 3.3, and Mask AR by 2.3. The most remarkable performance gain is observed on **SA-1B**. **We improve Box AR from 18.8 to 29.0 (relatively +54%) and improve Mask AR from 17.0 to 26.0 (relatively +53%).**

We gain improved recalls for objects of all scales, but *small and medium-sized objects relatively benefit more than large objects*. For instance, our $AR_S$ is 2.6× as CutLER's $AR_S$ on SA-1B. It is worth noting that detecting small objects is intrinsically harder than large objects – even though the

Table 1: Comparison of self-supervised object detection and instance segmentation methods on prevalent image datasets. We consider the average recall (AR) instead of average precision (AP) as the main metric, because valid detection of objects outside the categories defined by human annotations is penalized by AP. HASSOD significantly outperforms the previously best methods FreeSOLO [37] and CutLER [38] in terms of AR at all object scales (**S**mall, **M**edium, and **L**arge). To understand the extent of improvements, we also include results from state-of-the-art *supervised* model SAM [18]. HASSOD leads to a reduced gap between *fully self-supervised* models and *supervised* SAM. Notably, HASSOD only uses $1/5$ of training images and $1/12$ of training iterations as CutLER.

| Method | Box | | | | | Mask | | | | |
|---|---|---|---|---|---|---|---|---|---|---|
| | AR | $AR_S$ | $AR_M$ | $AR_L$ | AP | AR | $AR_S$ | $AR_M$ | $AR_L$ | AP |
| *Objects365* [27] | | | | | | | | | | |
| SAM [18] | 54.9 | 32.1 | 60.5 | 67.6 | 11.9 | | | | | |
| FreeSOLO [37] | 10.2 | 0.2 | 5.8 | 23.4 | 3.4 | No ground-truth mask | | | | |
| CutLER [38] | 35.8 | 17.6 | 36.1 | 50.5 | **11.5** | annotations in Objects365 | | | | |
| HASSOD (Ours) | **39.0** | **21.4** | **40.4** | **52.1** | 11.0 | | | | | |
| *LVIS* [11] | | | | | | | | | | |
| SAM [18] | 42.7 | 27.7 | 66.3 | 75.5 | 6.1 | 46.1 | 31.1 | 71.3 | 74.6 | 6.7 |
| FreeSOLO [37] | 6.4 | 0.3 | 9.7 | 34.6 | 1.9 | 5.9 | 0.2 | 9.2 | 31.7 | 1.9 |
| CutLER [38] | 23.6 | 13.1 | 36.2 | 55.6 | 4.5 | 20.2 | 11.3 | 31.1 | 46.2 | 3.6 |
| HASSOD (Ours) | **26.9** | **15.6** | **42.2** | **56.9** | **4.9** | **22.5** | **12.7** | **36.1** | **47.8** | **4.2** |
| *SA-1B* [18] | | | | | | | | | | |
| SAM [18] | 60.5 | 19.8 | 59.8 | 81.5 | 38.2 | 60.8 | 20.0 | 59.9 | 82.2 | 38.9 |
| FreeSOLO [37] | 2.4 | 0.0 | 0.1 | 7.4 | 1.5 | 2.2 | 0.0 | 0.2 | 6.9 | 1.5 |
| CutLER [38] | 18.8 | 5.1 | 14.6 | 32.8 | 9.0 | 17.0 | 4.9 | 13.9 | 28.5 | 7.8 |
| HASSOD (Ours) | **29.0** | **13.3** | **25.1** | **43.8** | **15.5** | **26.0** | **12.9** | **22.8** | **38.3** | **13.8** |

labels in SA-1B are produced by the same SAM model, when the bounding box prompts are no longer available, SAM can only reach a 20.0 Mask AR for small objects. Meanwhile, **we halve the performance gap between self-supervised CutLER and supervised SAM from 15.1 Mask $AR_S$ to 7.1 Mask $AR_S$.** By learning from hierarchical levels of object compositions, HASSOD more effectively captures small objects which are part of whole objects.

## 4.4 Qualitative Results

In this section, we analyze some qualitative results on images from LVIS. The visualization is shown in Figure 5. Qualitative results on other datasets are included in Appendix K.

As shown in the examples, our proposed HASSOD exhibits a more comprehensive coverage of all objects in complex scenes, compared with the previous state-of-the-art self-supervised object detection method CutLER [38]. This advantage originates from the pseudo-labels generated by our hierarchical adaptive clustering, which includes a proper number of candidate objects per image according to the image contents, rather than only focusing on a fixed number of objects. Furthermore, HASSOD can predict the hierarchical level of each detected object. Being a *fully self-supervised* model, HASSOD has surprisingly gained the *human-like ability* to comprehend the composition of objects. This ability leads to better interpretability and controllability: Users of HASSOD can understand the composition of each object detected by the model. Meanwhile, users can also control the segmentation granularity by selecting the predictions at the desired hierarchical level.

The qualitative results also show some limitations of HASSOD. By comparing the last two columns, it can be observed that HASSOD produces relatively fewer predictions for "subpart" objects. This is due to the distribution imbalance in the initial pseudo-labels, in which only about 10% objects are subparts. Also, the hierarchical levels learned by HASSOD are sometimes inconsistent with human perception. For example, instead of recognizing the person as a whole object in the last example image, HASSOD detects the upper and lower parts of the body as whole objects. Due to the lack of human supervision, the hierarchical prediction of HASSOD is not always aligned with humans. We further analyze this limitation in Appendix I.

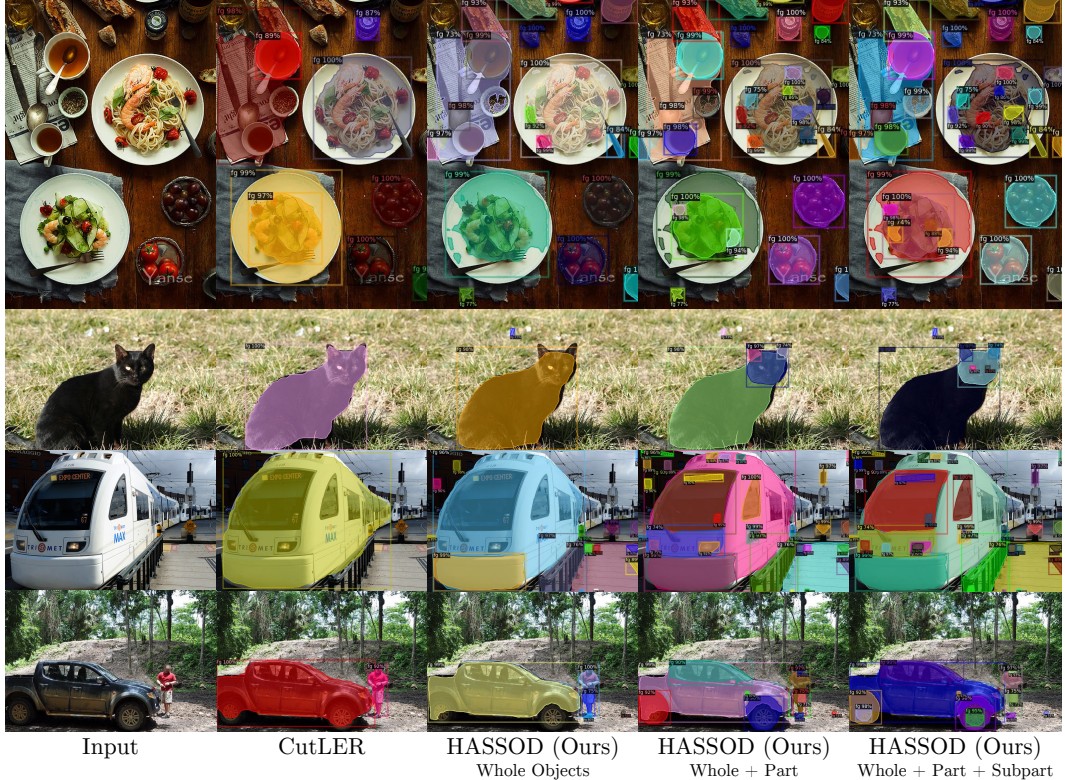

| Input | CutLER | HASSOD (Ours) Whole Objects | HASSOD (Ours) Whole + Part | HASSOD (Ours) Whole + Part + Subpart |

Figure 5: Qualitative results on LVIS images. Overall, our HASSOD successfully detects more objects compared with CutLER [38]. CutLER tends to detect only one or few prominent objects in the image, while HASSOD captures other objects as well (*e.g.*, bread in row 1, and traffic sign in row 3). Moreover, HASSOD learns the composition of objects (*e.g.*, cat-face-eye in row 2, and vehicle-wheel-tire in row 4), which is similar to human perception.

## 4.5 Ablation Study

In this section, we conduct an ablation study to understand the effects of each component in HASSOD. To evaluate the performance more robustly against a larger set of human-annotated object-level labels, we combine the annotations of MS-COCO [20] and LVIS [11] on the `val2017` split, because they are complementary to each other: LVIS uses the same images as MS-COCO, but labels more object classes. However, LVIS annotations are not as exhaustive as MS-COCO, meaning that objects within LVIS categories may not be labeled on all images. After combining the two sets of annotations and removing duplicates, there are around 20 object-level labels per image. We use this combined dataset for evaluation in all the ablation study experiments, unless otherwise specified.

**Quality of initial pseudo-labels.** We first examine how the design choices in our hierarchical adaptive clustering influence the quality of initial pseudo-labels. The results are summarized in Table 2. Each threshold $\theta_i^{\text{merge}} \in \{0.1, 0.2, 0.4\}$ leads to a different trade-off between the number of labels per image and the recall. A higher threshold stops the merging process earlier, and thus results in more pseudo-labels. With post-processing, we can improve pseudo-label quality by removing about half of the labels, and increase AP significantly without losing much AR. Finally, the ensemble of multiple merging thresholds $\theta_i^{\text{merge}} \in \{0.1, 0.2, 0.4\}$ brings the best overall pseudo-label quality. Appendices D, E, and F present more details regarding the choice of $\theta^{\text{merge}}$, computation costs, and ViT backbones in this stage.

**Effects of hierarchical level prediction and Mean Teacher.** After generating the initial pseudo-labels with hierarchical adaptive clustering, we train the object detector with several techniques, including hierarchical level prediction, Mean Teacher self-training, and adaptively adjusting learning targets. The contribution of each technique is summarized in Table 3. Each component brings an additional 0.3 - 0.5 Mask AR improvement, and when they function together, the best overall performance can be achieved.

Table 2: Ablation study on factors influencing the quality of initial pseudo-labels. The threshold $\theta^{\mathrm{merge}}$ controls the stopping criterion of the merging process, and affects AR and the number of pseudo-labels. Applying post-processing can remove low-quality pseudo-labels without decreasing AR by a large margin. Ensemble of multiple pseudo-label sources leads to the best overall quality.

| Method | $\theta^{\mathrm{merge}}$ | Post-Process | Labels per Img | AR | $AR_S$ | Mask $AR_M$ | $AR_L$ | AP |
|---|---|---|---|---|---|---|---|---|
| MaskCut [38] | – | ✓ | 1.85 | 3.5 | 0.0 | 2.0 | 20.0 | 1.5 |
| Hierarchical adaptive clustering (Ours) | 0.1 | | 5.33 | 4.4 | 0.8 | 5.1 | 16.6 | 1.2 |
| | 0.1 | ✓ | 2.58 | 4.1 | 0.6 | 5.1 | 15.6 | **1.8** |
| | 0.2 | | 8.36 | 5.5 | 1.2 | 7.0 | 18.9 | 1.3 |
| | 0.2 | ✓ | 4.20 | 5.3 | 0.9 | 7.0 | 18.4 | **1.8** |
| | 0.4 | | 23.33 | 7.9 | **2.1** | 12.0 | 21.7 | 0.7 |
| | 0.4 | ✓ | 11.61 | 7.8 | 1.7 | 12.1 | 22.1 | 1.3 |
| | ensemble | ✓ | 12.69 | **8.9** | 1.7 | **12.4** | **29.1** | 1.7 |

Table 3: Ablation study on factors influencing the training of the object detector. Hierarchical level prediction is introduced as an auxiliary task for self-supervision. Mean Teacher self-training replaces the vanilla multi-round self-training and brings a smoother and more efficient training process. The weights of two learning targets, initial pseudo-labels and teacher predictions, are adaptively adjusted to build a more effective curriculum. All the three key designs are combined for the best overall performance of HASSOD.

| Level Prediction | Mean Teacher | Adaptive Targets | AR | $AR_S$ | Mask $AR_M$ | $AR_L$ | AP |
|---|---|---|---|---|---|---|---|
| | | | 20.2 | 9.2 | 30.4 | 42.5 | 5.7 |
| ✓ | | | 20.6 | 9.7 | 30.1 | 43.9 | 6.1 |
| ✓ | ✓ | | 22.1 | 10.9 | **32.9** | 44.5 | 5.5 |
| ✓ | ✓ | ✓ | **22.4** | **11.3** | **32.9** | **45.0** | **6.3** |

**Improvement over initial pseudo-labels.** Although the initial pseudo-labels are produced by a frozen DINO backbone and they tend to be noisy and coarse, HASSOD is *not upper-bounded by* the quality of the fixed initial pseudo-labels or the pre-trained backbone. This is because of the following reasons: 1) While the initially discovered pseudo-labels are noisy, in the learning stage we train a detection model to learn common objects and their hierarchical relations for *enhanced generalization* to unseen images. By learning this detector, we boost the detection AR and AP from 8.9 and 1.7 (the last row in Table 2) to 20.6 and 6.1 (the second row in Table 3), respectively. Meanwhile, the pre-trained backbone features are adapted for the detection task in an end-to-end manner. 2) We further leverage Mean Teacher for continual self-enhancement, and gradually *minimize the negative impact of noisy initial pseudo-labels*. The evolving teacher detector and its features provide improved pseudo-labels to the student. Notably, we can directly *read out* the predictions from the teacher as the refined hierarchical pseudo-labels, instead of inefficiently running the clustering algorithm using the enhanced backbone. Consequently, we further improve the detection AR and AP to 22.4 and 6.3 (the last row in Table 3), respectively.

## 5 Conclusion

We present Hierarchical Adaptive Self-Supervised Object Detection (HASSOD), an approach inspired by human visual perception that learns to detect objects and understand object composition in a self-supervised manner. HASSOD uses a hierarchical adaptive clustering strategy to propose a varying number of objects per image, and learns hierarchical levels of objects by analyzing geometric relations between objects. Mean Teacher self-training with adaptive targets facilitates the detector training process with smooth learning objectives and improved training efficiency. Empirical evaluation on recent large-scale image datasets Objects365, LVIS, and SA-1B demonstrates our significant improvement over prior self-supervised detectors. We detail the limitations and broader impacts of HASSOD in Appendix I.

## Acknowledgments

This work was supported in part by the IBM-Illinois Discovery Accelerator Institute, NSF Grant #2106825, NIFA Award #2020-67021-32799, the Jump ARCHES endowment through the Health Care Engineering Systems Center, the National Center for Supercomputing Applications (NCSA) at the University of Illinois at Urbana-Champaign through the NCSA Fellows program, the Illinois-Insper Partnership, and the Amazon Research Award. This work used NVIDIA GPUs at NCSA Delta through allocations CIS220014, CIS230012, and CIS230013 from the Advanced Cyberinfrastructure Coordination Ecosystem: Services & Support (ACCESS) program [3], which is supported by NSF Grants #2138259, #2138286, #2138307, #2137603, and #2138296.

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

# Appendix

In this appendix, Section A first explains the deficiency of traditional MS-COCO AP evaluation in the self-supervised setting and reasons for adopting AR on more class-extensively-annotated datasets like LVIS. Sections B and C address potential concerns about unfair comparison with CutLER [38] regarding the training data and the detector architecture. Sections D, E, and F present additional ablation study regarding the threshold choice, computation cost, and patch size of the ViT backbone in our hierarchical adaptive clustering algorithm. Section G studies different behaviors of SAM and our proposed HASSOD in detecting parts of objects. Sections H and K present more comprehensive quantitative and qualitative evaluation results for completeness. Section I provides the failure cases of HASSOD and analyzes its current limitations. Section J describes the detailed hyper-parameter and implementation setup.

## A  Deficiency of MS-COCO AP Evaluation in Self-Supervised Object Detection

Traditionally, the Average Precision (AP) metric on the MS-COCO dataset [20] has been a gold standard for assessing the performance of *supervised* object detection and instance segmentation models, which are trained and evaluated on a fixed set of object categories pre-defined by human annotators. However, *we find this metric misleading in the context of self-supervised object detection*, where class labels are not available to the model and *class-agnostic* predictions are necessary. In this work, we advocate for evaluating Average Recall (AR) on datasets with extensive class annotations (*e.g.*, LVIS [11]) as a more reliable metric for comparing different methods. In this section, we discuss the inherent deficiencies of MS-COCO AP evaluation using an illustrative example.

To demonstrate this problem objectively, we compare two previous methods, CutLER [38] and SAM [18]. Both models function as class-agnostic object detectors, but SAM, trained with human supervision, evidently outperforms CutLER in terms of detecting and segmenting objects, as depicted in Figure 6. Surprisingly, when evaluated on MS-COCO annotations, SAM attains a mere 6.7 AP, which is approximately half of CutLER's 12.3 AP. In this case, the AP metric is contradicting with the observed performance of the two models in terms of accurately detecting and segmenting as many objects as possible.

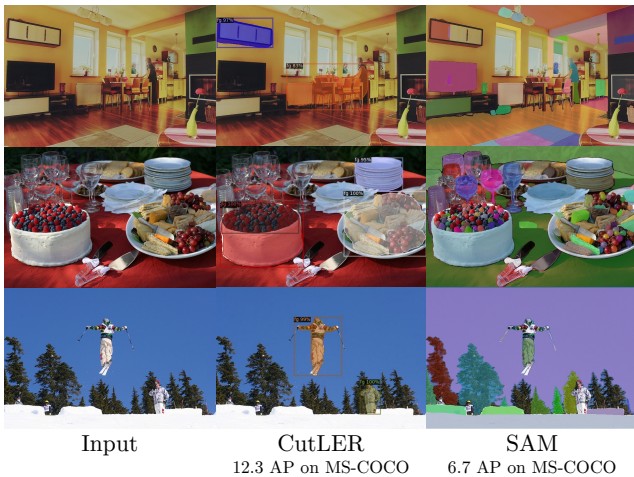

|  Input  |  CutLER  |  SAM  |
|---------|----------|-------|
|         | 12.3 AP on MS-COCO | 6.7 AP on MS-COCO |

Figure 6: MS-COCO average precision (AP) evaluation may not accurately reflect the performance of *class-agnostic* object detectors. We compare two previous class-agnostic object detection/segmentation models, CutLER [38] and SAM [18]. Despite SAM, a supervised model, detecting more objects with superior localization than CutLER, it achieves only half the AP of CutLER.

The primary reason for this discrepancy lies in the annotations of MS-COCO. MS-COCO labels only 80 object categories, and model predictions are considered as true positives only if they fall within

these categories. Consequently, correct predictions for objects outside the 80 categories are unjustly deemed false positives and penalized by the AP metric, contradicting the objective of class-agnostic object detection and revealing the shortcomings of AP evaluation.

To correct these deficiencies in the evaluation metric, we need two changes: 1) *Adopt a dataset with comprehensive annotations that include as many object categories as possible.* If we substitute MS-COCO ground-truth annotations with LVIS, which labels over 1,200 object categories despite using the same images, the AP comparison is reversed: CutLER scores 4.5 AP on LVIS, while SAM attains a higher 6.1 AP. 2) *Replace AP with the AR metric.* Even with LVIS annotations, not all object categories are labeled in every image, resulting in certain valid object detection predictions still being penalized. Conversely, AR does not penalize such predictions and exhibits a more pronounced difference between CutLER and SAM (23.6 AR *vs.* 42.7 AR).

In summary, we advocate for AR on class-extensively-annotated datasets as the primary comparison metric, which can more accurately reflect the actual performance of class-agnostic object detectors with reduced bias. This choice of metric aligns with prior work on open-world detection [1, 17], segmentation [16, 36], and tracking [21] as well.

## B   Comparison of CutLER and HASSOD with Equal Training Data

As described in the main paper, HASSOD utilizes a ResNet-50 backbone initialized with DINO [5] weights pre-trained in a self-supervised manner on ImageNet [7], while subsequent training is conducted on MS-COCO [20] images. We choose MS-COCO for its compact size and richness in objects per image. Due to limited computation resources, we are unable to perform HASSOD training on ImageNet, and we leave large-scale training as one interesting future direction. This distinction in the training dataset may lead to concerns regarding the fairness of comparison with prior work, such as CutLER [38], which trains exclusively on ImageNet data. However, it is important to note that using both ImageNet and MS-COCO data in HASSOD does not grant HASSOD additional benefit compared with CutLER for two main reasons: 1) The two datasets are leveraged in separate stages and not in a blended manner. Specifically, ImageNet data are only used in the DINO pre-training stage, while MS-COCO is employed for detector training. 2) ImageNet contains approximately $5\times$ as many images as MS-COCO. Thus, if ImageNet is employed in the detector training stage, as is the case with CutLER, this would actually lead to a stronger detector.

Table 4: Comparison between CutLER [38] and HASSOD concerning the *training image dataset*. Although both CutLER and HASSOD leverage DINO [5] weights pre-trained on ImageNet, employing MS-COCO images in the detector training stage does not lead to superior performance. CutLER trained on ImageNet surpasses CutLER trained on MS-COCO across all metrics. Simultaneously, HASSOD outperforms both CutLER models, despite using MS-COCO training data and requiring fewer training iterations.

| Method | Training Images | Training Iterations | AR | $AR_S$ | Mask $AR_M$ | $AR_L$ | AP |
|---|---|---|---|---|---|---|---|
| CutLER [38] | MS-COCO [20] | 160,000 | 17.3 | 6.2 | 24.2 | 46.1 | 5.8 |
| CutLER [38] | ImageNet [7] | 160,000 | 18.8 | 7.2 | 27.6 | **46.6** | 6.2 |
| HASSOD (Ours) | MS-COCO [20] | 40,000 | **22.4** | **11.3** | **32.9** | 45.0 | **6.3** |

In order to address these concerns more effectively and ensure an *equal usage of training data*, we conduct an additional experiment. Specifically, we train a Cascade Mask R-CNN [4] detector using the CutLER approach on MS-COCO images for one round (160,000 iterations), with the ResNet-50 backbone initialized with DINO pre-trained weights. We adopt CutLER's original implementation, with the sole modification being the use of MS-COCO images for training. We compare this model with a CutLER model trained on ImageNet for one round and our HASSOD model trained on MS-COCO. The evaluation is conducted against MS-COCO+LVIS annotations, as described in Section 4.5. The results are presented in Table 4.

By comparing the two CutLER models trained with MS-COCO and ImageNet, we observe that utilizing distinct datasets for backbone pre-training and detector training does not universally improve performance. The CutLER model, exclusively trained on ImageNet, surpasses its counterpart that uses

ImageNet for DINO pre-training and MS-COCO for detector training, exhibiting gains of 1.5 Mask AR and 0.4 Mask AP. Meanwhile, HASSOD, *despite being trained on the smaller MS-COCO dataset and for a shorter duration, outperforms CutLER by 3.6 Mask AR*. This remarkable performance is attributed to HASSOD's comprehensive object coverage in images and its efficient usage of training data.

## C  Additional Results on Mask R-CNN

In HASSOD, we train a Cascade Mask R-CNN [4] object detection and instance segmentation model. We choose this architecture following CutLER [38] and we ensure a fair comparison with this prior work. In fact, HASSOD can be applied on different detector architectures. As an example, we also train a Mask R-CNN [14] and compare its LVIS performance with CutLER in Table 5. With the same detector architecture, HASSOD produces a better detector than CutLER. More impressively, *our Mask R-CNN (weaker architecture) outperforms CutLER's Cascade Mask R-CNN (stronger architecture) on LVIS*, highlighting the advantage of our approach.

Table 5: Comparison between CutLER [38] and HASSOD with different detector architectures on the LVIS dataset. When training models of the same architecture, HASSOD outperforms CutLER (row 1 *vs*. 2, row 3 *vs*. 4). Notably, HASSOD has a stronger AR even with a weaker architecture (row 2 *vs*. 3) as compared with CutLER.

| Architecture | Method | Box | | | | | Mask | | | | |
|---|---|---|---|---|---|---|---|---|---|---|---|
| | | AR | $AR_S$ | $AR_M$ | $AR_L$ | AP | AR | $AR_S$ | $AR_M$ | $AR_L$ | AP |
| Mask R-CNN | CutLER [38] | 20.7 | 10.4 | 33.3 | **52.0** | 4.1 | 18.5 | 9.6 | 29.1 | 44.9 | 3.4 |
| | HASSOD (Ours) | **23.8** | **13.5** | **38.3** | 50.9 | **4.3** | **21.5** | **11.8** | **35.1** | **46.6** | **4.1** |
| Cas. Mask R-CNN | CutLER [38] | 23.6 | 13.1 | 36.2 | 55.6 | 4.5 | 20.2 | 11.3 | 31.1 | 46.2 | 3.6 |
| | HASSOD (Ours) | **26.9** | **15.6** | **42.2** | **56.9** | **4.9** | **22.5** | **12.7** | **36.1** | **47.8** | **4.2** |

## D  Choosing Thresholds for Hierarchical Adaptive Clustering

When determining the merging thresholds $\{\theta_i^{\text{merge}}\}$, we mainly consider our computational constraints and empirical observations. The decision of merging thresholds is *not* made based on validation performance (Table 2), ensuring that HASSOD is fully self-supervised.

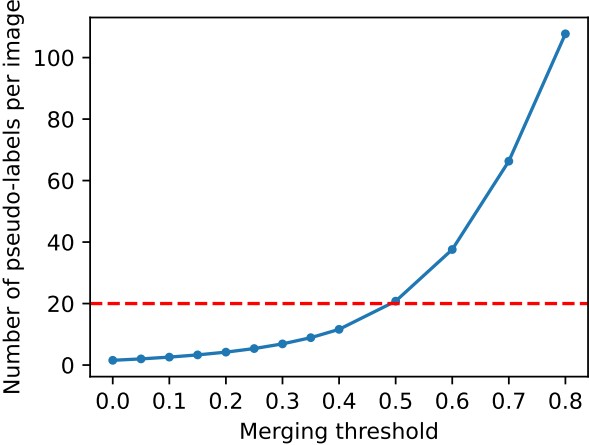

Figure 7: Relation between the number of pseudo-labels per image after post-processing (y-axis) and the merging threshold $\theta^{\text{merge}}$ (x-axis). When $\theta^{\text{merge}} \geq 0.5$, the number of pseudo-labeled masks grows rapidly. Empirically, we find that when the number of masks per image exceeds 20, generating, loading, and transforming such pseudo-labels becomes a major bottleneck in HASSOD. Therefore, we choose three thresholds $\{\theta_i^{\text{merge}}\} = \{0.1, 0.2, 0.4\}$, mainly guided by a computational consideration.

**Guidance by number of pseudo-masks.** Our choice for $\{\theta_i^{\mathrm{merge}}\}$ is primarily guided by the number of pseudo-label masks produced per image. Figure 7 shows the relationship between the number of masks per image and different thresholds. When $\theta^{\mathrm{merge}} \geq 0.5$, the number of masks per image escalates rapidly. This steep increase incurs significant computational costs, both during the initial generation of pseudo-labels and the subsequent data loading and pre-processing procedures during model training. To strike a balance between computational efficiency and the desired mask granularity, thresholds of $\{\theta_i^{\mathrm{merge}}\} = \{0.1, 0.2, 0.4\}$ are chosen.

Table 6: Generalizability of the merging thresholds $\{\theta_i^{\mathrm{merge}}\}$. With the same merging threshold, the number of generated pseudo-labels is not significantly changing across different image datasets.

| $\theta^{\mathrm{merge}}$ | Number of pseudo-labels per image | | |
|---|---|---|---|
| | MS-COCO [20] | Objects365 [27] | SA-1B [18] |
| 0.1 | 2.58 | 3.49 | 2.91 |
| 0.2 | 4.20 | 5.78 | 4.88 |
| 0.4 | 11.61 | 12.15 | 12.70 |

**Threshold generalization across datasets.** Another noteworthy observation is the generalizability of these thresholds across various datasets. In Table 6, we present the number of generated pseudo-labels per image on three datasets. With the merging threshold $\theta^{\mathrm{merge}}$ fixed, the number of generated labels is relatively stable, regardless of the source image dataset. Therefore, our pre-set thresholds are generalizable and require no further tuning when transferred to other datasets. Meanwhile, our detection model was trained on MS-COCO images with pseudo-labels generated using our pre-set $\{\theta_i^{\mathrm{merge}}\}$, and could generalize well to other datasets in a zero-shot manner, as shown in Table 1. This fact shows that the thresholds $\{\theta_i^{\mathrm{merge}}\}$ are effective regardless of evaluation datasets.

# E Computational Costs in Hierarchical Adaptive Clustering

We adopt the hierarchical adaptive clustering strategy to generate our initial pseudo-labels. In this section, we provide additional details regarding computation costs in this procedure.

As shown in Figure 3, the hierarchical adaptive clustering contains four steps "merge," "post-process," "ensemble," and "split." Among them, the "merge" step accounts for the major computation costs. We can analyze its time complexity: Suppose we have $n$ patches in the beginning, then there are at most $n$ merging steps before stopping. Each merging step requires retrieving the most similar pair of adjacent regions. The collection of adjacent pairs has size at most $O(n)$, and each operation requires time $O(\log n)$ if this collection is organized as a balanced binary tree. Therefore, the merging process has time complexity $O(n \log n)$. In our practice, the input image has resolution $480 \times 480$, so $n = \frac{480}{8} \times \frac{480}{8} = 3,600$. This time complexity is affordable.

More concretely, we list the time costs in the hierarchical adaptive clustering on our computation platform in Table 7. Since the procedure is learning-free, we can *parallelize* the processing of images using more than one worker. On our computation nodes equipped with 4 NVIDIA A100 GPUs, we can reduce the processing time to 1.59 sec/image. With 4 such nodes, we can complete the pseudo-label generation for MS-COCO `train` and `unlabeled` splits (0.24 million images) in $\frac{1.59 \times 0.24 \times 10^6}{4 \times 86400} \approx 1$ day.

Table 7: Time costs in the steps of the hierarchical adaptive clustering. Notably, with multiple parallel workers, we can reduce the total processing time of MS-COCO [20] images to one day.

| Step | Time Cost (sec/image) | Workers | Parallelized Cost (sec/image) |
|---|---|---|---|
| Merge and Post-Process | 11.7 | 8 | 1.46 |
| Ensemble and Split | 2.1 | 16 | 0.13 |
| Total | 13.8 | - | 1.59 |

# F Impact of Patch Size in Hierarchical Adaptive Clustering

We use the DINO [5] pre-trained ViT-B/8 backbone to generate the initial pseudo-labels through our hierarchical adaptive clustering. We observe that the small patch size $8 \times 8$ leads to better pseudo-label quality. In Table 8, we compare the mask quality of pseudo-labels generated by ViT-B/8 (patch size $8 \times 8$) *vs.* ViT-B/16 (patch size $16 \times 16$). For a fair comparison, we use $480 \times 480$ input resolution for ViT-B/8 and $960 \times 960$ for ViT-B/16, so that they have the same number of initial patches. With the same merging threshold $\theta^{\mathrm{merge}}$, ViT-B/16 leads to slightly fewer labels per image, and the quality is significantly worse than ViT-B/8, especially for small and medium objects. Therefore, we apply ViT-B/8 in our experiments for its localized visual features and subsequent high-quality pseudo-labels.

Table 8: Comparison between DINO ViT backbones with different patch sizes $8 \times 8$ and $16 \times 16$. The backbone with the smaller patch size leads to higher-quality initial pseudo-labels in the hierarchical adaptive clustering procedure.

| DINO Backbone | $\theta^{\mathrm{merge}}$ | Labels per Img | AR | $AR_S$ | Mask $AR_M$ | $AR_L$ | AP |
|---|---|---|---|---|---|---|---|
| ViT-B/8 | 0.1 | 2.58 | 4.1 | 0.6 | 5.1 | 15.6 | 1.8 |
|  | 0.2 | 4.20 | 5.3 | 0.9 | 7.0 | 18.4 | 1.8 |
|  | 0.4 | 11.61 | 7.8 | 1.7 | 12.1 | 22.1 | 1.3 |
| ViT-B/16 | 0.1 | 1.97 | 3.0 | 0.3 | 2.3 | 14.6 | 1.1 |
|  | 0.2 | 3.19 | 3.8 | 0.4 | 3.3 | 17.4 | 1.2 |
|  | 0.4 | 10.15 | 5.7 | 0.9 | 6.6 | 21.7 | 1.3 |

# G Different Behaviors of SAM and HASSOD in Object Part Detection

Segment Anything Model (SAM) [18], a supervised segmentation model, has demonstrated its ability in creating high-quality, fine-grained segmentation of images. Directly comparing the performance between SAM and HASSOD would be unbalanced, considering SAM requires extensive human-labeled images for training, while HASSOD operates entirely under self-supervision. Despite this, it remains intriguing to understand their different behaviors. In this section, we delve into a qualitative comparison between SAM and our proposed HASSOD approach, paying particular attention to their respective abilities to detect constituent parts of whole objects. Figure 8 presents a visualization for this comparison.

Both SAM and HASSOD can perform fine-grained segmentation within complex scenes, successfully detecting individual object parts that constitute a whole entity. However, a clear distinction arises in their respective approaches towards the segmentation of these object parts. For scenes in which objects follow a grid pattern, where the whole entity is partitioned by regular boundary lines into semantically similar pieces, SAM tends to perceive each grid as a distinct object. Conversely, HASSOD adheres to a holistic perspective for such cases, grouping the grid parts together even at the subpart level, due to their semantically analogous features.

In particular, HASSOD excels at distinguishing object parts that contain different contents within whole objects. This skill of HASSOD is especially advantageous in certain real-world applications. For example, in medical imaging, HASSOD's fine-grained segmentation can potentially assist in identifying and separating different tissues or anatomical structures within a scan. Additionally, in the manufacturing industry, HASSOD could be used in quality control to check and compare individual components of an assembled product. While both segmentation strategies of SAM and HASSOD are valid, HASSOD's unique ability to distinguish semantically different parts within objects provides distinct advantages in a wide range of practical scenarios. Comprehensively quantifying and analyzing such an ability is crucial for advancement of self-supervised object detection and segmentation approaches, and we consider it as an important future direction.

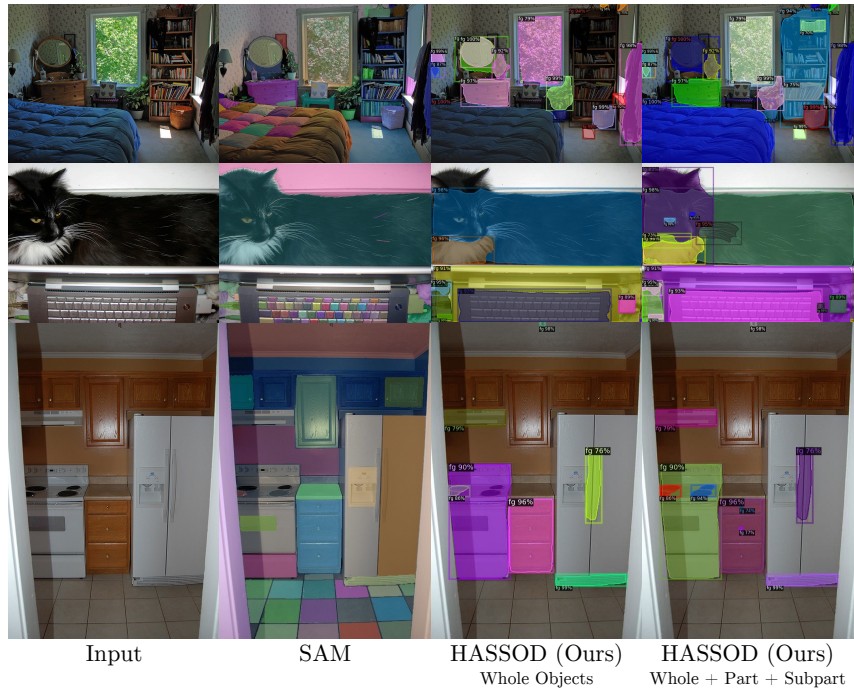

| Input | SAM | HASSOD (Ours)
Whole Objects | HASSOD (Ours)
Whole + Part + Subpart |

Figure 8: Analysis of different behaviors in object part detection between *supervised* SAM [18] and *self-supervised* HASSOD through qualitative visualization. While both models generate fine-grained segmentation, they exhibit distinct preferences concerning object parts. SAM inclines towards separating objects following a grid pattern (*e.g.*, comforters, keyboards, tiles), whereas HASSOD discriminates objects into semantically diverse parts (*e.g.*, eyes and beard of a cat, handle and button of a cabinet).

## H  Additional Evaluation Results for Completeness

As an addition to Table 1 in the main paper, we present a comprehensive evaluation of HASSOD on the Objects365 [27], LVIS [11], and SA-1B [18] datasets in Table 9. For a robust measure of performance, we incorporate the standard deviation, estimated from three independently trained models.

Table 9: Comprehensive evaluation results of HASSOD across three datasets. We set the maximum number of predictions per image to 1,000. The number superscript on AR (*e.g.*, $AR^{10}$) denotes the number of most confident predictions taken into account when computing AR. The subscript on AP (*e.g.*, $AP_{50}$) represents the Intersection-over-Union (IoU) threshold utilized when matching predictions with ground truth labels. Size-specific metrics for small, medium, and large objects are indicated by the subscripts $S$, $M$, and $L$, respectively. We include the standard deviation, estimated from three independent runs.

| Dataset | $AR^{10}$ | $AR^{100}$ | $AR^{1000}$ | $AR_S$ | $AR_M$ | $AR_L$ | $AP$ | $AP_{50}$ | $AP_{75}$ | $AP_S$ | $AP_M$ | $AP_L$ |
|---|---|---|---|---|---|---|---|---|---|---|---|---|
| | \multicolumn{12}{c}{Box *(Object Detection)*} | | | | | | | | | | | |
| Objects365 [27] | 15.20 | 36.63 | 39.03 | 21.40 | 40.43 | 52.10 | 10.97 | 20.33 | 10.27 | 2.90 | 10.37 | 19.47 |
| | ±0.10 | ±0.06 | ±0.06 | ±0.10 | ±0.12 | ±0.17 | ±0.15 | ±0.31 | ±0.15 | ±0.10 | ±0.25 | ±0.25 |
| LVIS [11] | 10.58 | 25.01 | 26.87 | 15.64 | 42.22 | 56.90 | 4.94 | 9.03 | 4.75 | 2.82 | 7.90 | 12.19 |
| | ±0.07 | ±0.02 | ±0.03 | ±0.05 | ±0.08 | ±0.17 | ±0.09 | ±0.09 | ±0.10 | ±0.01 | ±0.14 | ±0.20 |
| SA-1B [18] | 5.52 | 23.92 | 29.02 | 13.34 | 25.12 | 43.79 | 15.47 | 26.20 | 15.90 | 5.39 | 15.06 | 21.81 |
| | ±0.02 | ±0.03 | ±0.18 | ±0.27 | ±0.21 | ±0.12 | ±0.08 | ±0.09 | ±0.03 | ±0.06 | ±0.13 | ±0.08 |
| | \multicolumn{12}{c}{Mask *(Instance Segmentation)*} | | | | | | | | | | | |
| LVIS [11] | 9.69 | 21.11 | 22.50 | 12.68 | 36.14 | 47.79 | 4.21 | 7.99 | 3.95 | 1.88 | 6.96 | 13.67 |
| | ±0.06 | ±0.05 | ±0.01 | ±0.01 | ±0.11 | ±0.17 | ±0.20 | ±0.25 | ±0.24 | ±0.19 | ±0.21 | ±0.18 |
| SA-1B [18] | 5.27 | 21.62 | 25.99 | 12.90 | 22.76 | 38.33 | 13.85 | 24.78 | 13.67 | 4.09 | 13.45 | 20.36 |
| | ±0.01 | ±0.07 | ±0.22 | ±0.37 | ±0.24 | ±0.21 | ±0.09 | ±0.08 | ±0.11 | ±0.04 | ±0.10 | ±0.07 |

# I Limitations and Broader Impacts

**Limitations.** Due to the self-supervised nature of HASSOD, the learned object hierarchical levels may not be perfectly aligned with human perception. This mismatch may lead to over-segment or under-segment of objects in real-world applications.

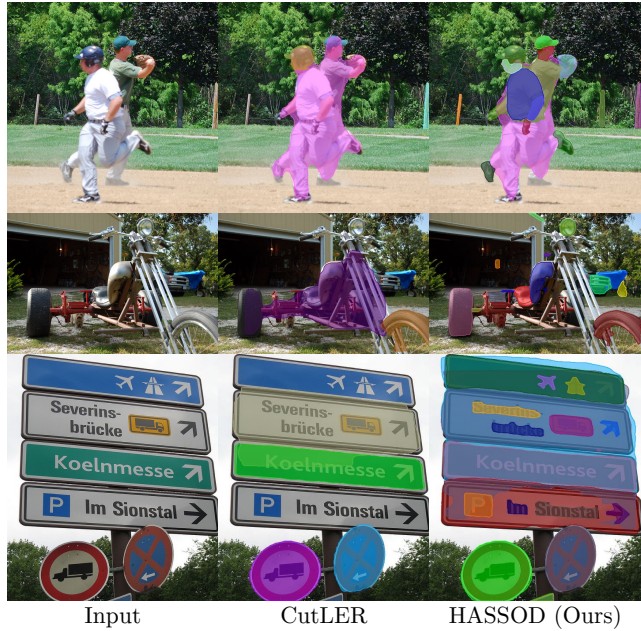

Input        CutLER      HASSOD (Ours)

Figure 9: Failure cases of HASSOD. Top: Overlapping and similar objects are hard to distinguish. Middle: The whole object comprising diverse parts is not identified. Bottom: Text contents are not precisely localized.

We observe that HASSOD performs unsatisfactorily in certain scenes due to this lack of human supervision. Figure 9 provides a visualization of the failure cases. In the first row, HASSOD mistakenly treats the lower parts of the two players as a single coherent object. Due to their similar color and texture, multiple overlapping instances of the same class can sometimes be perceived as one object. In the second row, HASSOD fails to predict a mask that encompasses the entire motorcycle. This object consists of highly heterogeneous parts, making it challenging to recognize them as components of a single entity. In the third row, HASSOD fails to detect the text "Koelnmesse," and the boundaries for other text are not clear. We also observe that such errors are not unique to HASSOD; they appear in prior self-supervised object detection methods like CutLER as well. We believe that further human supervision would be necessary for correcting these mistakes.

**Broader impacts.** Detecting object parts with self-supervision may be beneficial to real-world applications including robotic manipulation and inspection. As a general object detection method, we share risks associated with applying recognition models such as abuse of surveillance systems.

# J Hyper-Parameters and Implementation Details

In the initial pseudo-label generation process, we use a frozen ViT-B/8 model [8] pre-trained on unlabeled ImageNet [7] by DINO [5], a self-supervised representation learning method, to extract visual features of `train` and `unlabeled` images in MS-COCO [20]. Following prior work Cut-LER [38], we resize the resolution of each image to $480 \times 480$, leading to $60 \times 60$ patches as initial regions. The merging process stops at three thresholds $\theta_1^{\text{merge}} = 0.4, \theta_2^{\text{merge}} = 0.2, \theta_3^{\text{merge}} = 0.1$, and results from these three thresholds are ensembled after post-processing. The post-processing steps include Conditional Random Field (CRF) [19] and filling the holes in each mask. We also filter out low-quality masks that 1) have an Intersection-over-Union (IoU) smaller than 0.5 before and after CRF, 2) are smaller than 100 pixels, or 3) contain more than two corners of the image (which are likely

background). These post-processing steps are also used in prior work including CutLER [38]. After ensembling results from the three thresholds $\{\theta_i^{\mathrm{merge}}\}_{i=1}^3$, we identify the hierarchical levels of each object mask based on the coverage relation analysis. The coverage threshold is set to $\theta^{\mathrm{cover}}\% = 90\%$.

In the detector training stage, we train a Cascade Mask R-CNN [4] with a ResNet-50 [13] backbone on the same MS-COCO [20] images. The backbone is initialized from DINO [5] self-supervised pre-training. Our hyper-parameter setting for Mean Teacher mostly follows the practice of Unbiased Teacher [22]. The whole training process starts with a "burn-in" stage, during which the student model is only trained on the initial pseudo-labels with a fixed learning rate 0.01 and fixed loss weights. After the burn-in stage, the teacher model is introduced, and we gradually adjust the learning rate from 0.01 to 0, the loss weight in the label-to-student branch from 1.0 to 0.0, and the loss weight in the teacher-to-student branch from 2.0 to 3.0, all following a cosine schedule. The whole training process spans 40,000 iterations with a batch size of 16 images. The training is performed on $4\times$ NVIDIA A100 GPUs. Our code is developed based on `PyTorch` [24] and `Detectron2` [40].

## K    Additional Qualitative Results

In this section, we present visualization of detection results by CutLER [38] and HASSOD on Objects365 [27] in Figure 10 and SA-1B [18] in Figure 11 (see next pages). Qualitative results on LVIS has been included in the main paper, Figure 5.

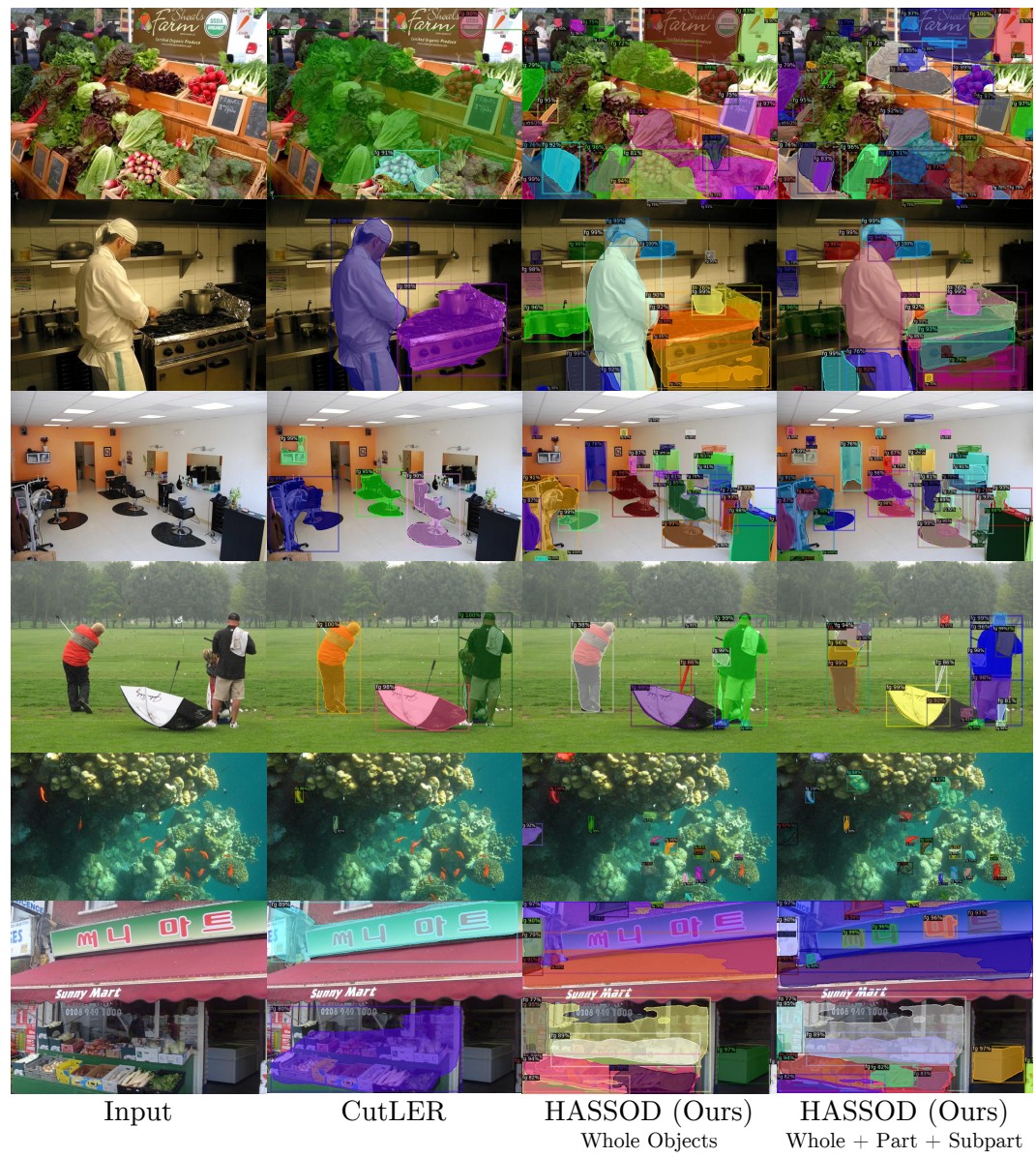

| Input | CutLER | HASSOD (Ours)
Whole Objects | HASSOD (Ours)
Whole + Part + Subpart |
|---|---|---|---|

Figure 10: Qualitative results on Objects365 [27]. In each row, we show detection results of prior state-of-the-art self-supervised object detector CutLER, whole objects predicted by HASSOD, and all object predicted by HASSOD.

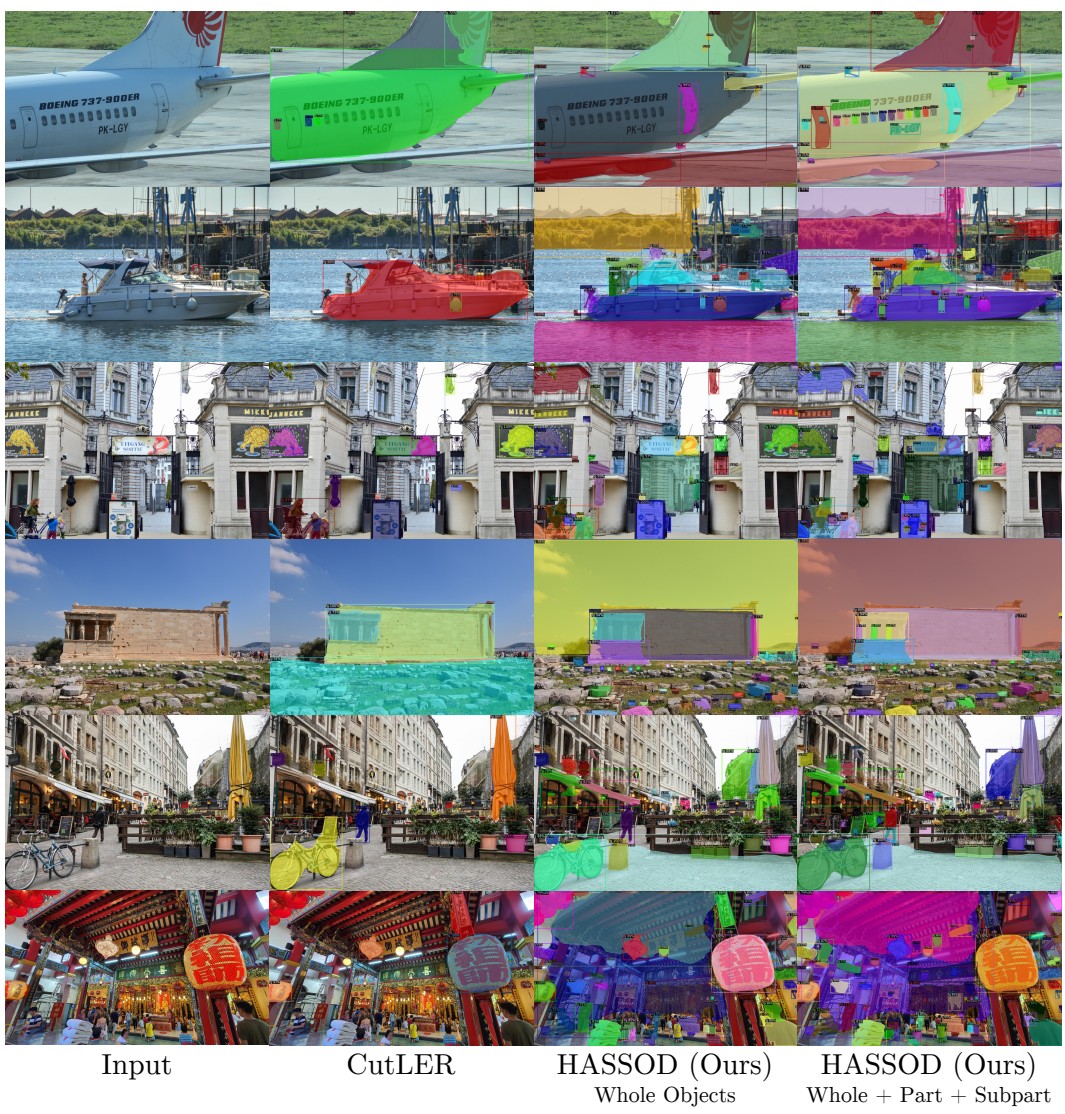

| Input | CutLER | HASSOD (Ours)
Whole Objects | HASSOD (Ours)
Whole + Part + Subpart |

Figure 11: Qualitative results on SA-1B [18]. In each row, we show detection results of prior state-of-the-art self-supervised object detector CutLER, whole objects predicted by HASSOD, and all object predicted by HASSOD.

