# OpenReview forum: "HASSOD: Hierarchical Adaptive Self-Supervised Object Detection"
_NeurIPS.cc/2023/Conference — NeurIPS 2023 poster_

### Official Review · Reviewer_wiMX · 2023-06-27

**Soundness:** 3 good
**Presentation:** 2 fair
**Contribution:** 2 fair
**Rating:** 5
**Confidence:** 5

**Summary:**

The paper proposes a self-supervised object detection method (in fact, it can also do instance segmentation). This is based on an idea of bottom-up merge by strong feature representation like from DINO feature. The image patch grouping is hierarchical because of setting different stopping thresholds. This makes the model interpretable. Overall, the idea is simple and straightforward. The model can be trained by teacher-student framework. Experiments show better results than recent work MaskCut.

**Strengths:**

- the bottom-up merging method to solve the self-supervised object detection might be new.
- Figure 1 and 2 are good to understand the method.
- experiment results look good.

**Weaknesses:**

- The writing is bad. Many critical details are omitted which makes it super hard to under the method.
- Some inconsistencies in the paper, needs to clarify.

**Questions:**

Some critical details are missing in the paper and they need to be clarified. 1) What is learned, the whole DINO backbone or just a head network? 2) For the Mean-Teacher, what are the losses used to supervise the student networks training?

The grouping process in section 3 is iterative and seems to be too slow. What is the time complexity?

Line 138, how to ensemble?

Line 195, in section 3, looks like only the feature representation is used, why train a Cascade mask rcnn here? Obviously, Cascade mask rcnn is not self-supervised.

Line 233 and 234 are inconsistent with section 4.2. What exactly Objects365, Lvis, SAM are for evaluation or training?

Lvis should have a LVIS-rare benchmark.

Whole, part, subpart should be given rigorous definitions.

For the limitations 2) and 3) in the second paragraph, I don't understand how they are solved after reading the whole paper.

**Limitations:**

The discussion at the end of the paper is good.

---

> ### Author Rebuttal · Authors · 2023-08-10
>
> We appreciate the detailed feedback you provided for our submission. We are encouraged by your acknowledgement that our method is “new”, figures are “good to understand”, and “experiment results look good”. We provide the following clarifications in response to your concerns:
>
> 1. Clarification on Learned Network and Mean Teacher
>
> - We learn the weights in both the DINO backbone and the detection head network. To be precise, our approach adopts a two-stage discover-and-learn procedure, and all weight learning happens in the second stage. Please see more details in the general response.
>
> - In Mean Teacher, the student network “receives supervision from two sources simultaneously” (Line 172): One source is the teacher’s predictions, and the other is the pseudo-labels generated in the first stage. The training losses are the *standard object detection losses*, including classification loss, bounding box regression loss, and mask prediction loss. Formally, the training loss for the student is $L_\text{student}=\alpha_\text{teacher}L_\text{det}(\tilde y_\text{student}, \tilde y_\text{teacher})+\alpha_\text{pseudo}L_\text{det}(\tilde y_\text{student}, y_\text{pseudo})$, where $\tilde y_\text{student}$ is the student's prediction for the given image, $\tilde y_\text{teacher}$ is the teacher's prediction, $y_\text{pseudo}$ is the pseudo-label, and $L_\text{det}$ is the standard detection loss mentioned above. $\alpha_\text{teacher}, \alpha_\text{pseudo}$ are loss weights dynamically adjusted during training, where $\alpha_\text{teacher}$ linearly increases while $\alpha_\text{pseudo}$ is linearly decreasing, as introduced in Line 183-186.
>
> 2. Time Complexity for Grouping Process
>
> Suppose we have $n$ patches in the beginning, then the time complexity of the grouping process is $O(n\log n)$: We have at most $n$ merging steps before stopping. Each merging step requires retrieving the most similar pair of adjacent regions. The collection of adjacent pairs has size at most $O(n)$, and each operation requires $O(\log n)$ time if this collection is organized as a balanced binary tree. Therefore, the grouping process has time complexity $O(n\log n)$. In our practice, the input image has resolution $480\times 480$, so $n=60\times 60=3600$. This time complexity is affordable. For more statistics, please check the general response.
>
> 3. Clarification on Ensemble
>
> In Line 138, “ensemble” means to aggregate all the pseudo-labels generated from different merging thresholds $\theta_i^\text{merge}$.
>
> 4. Cascade Mask R-CNN Training
>
> As explained above, our method takes a two-stage procedure. We train a Cascade Mask R-CNN in the second stage, using the pseudo-labels generated in the first stage. Since the pseudo-labels are generated from self-supervised features, our approach is still *fully self-supervised*.
>
> 5. Ambiguity in Dataset Usage
>
> Thank you for pointing out the ambiguity. We should parse the sentence as “We evaluate ... (Cascade Mask R-CNN trained by HASSOD) on Objects365, LVIS, and SA-1B”, rather than “We evaluate ... Cascade Mask R-CNN (trained by HASSOD on Objects365, LVIS, and SA-1B)”.
>
> To clarify, the Cascade Mask R-CNN learned by HASSOD is exclusively trained on MS-COCO images. The sentence in question means that after training, we evaluate the performance of this model on the Objects365, LVIS, and SA-1B datasets in a *zero-shot* manner. The model is not further trained on these datasets.
>
> 6. LVIS-rare Benchmark
>
> Our main objective is to identify **every object** present in scene images without distinguishing based on classes. We train and evaluate our model as a class-agnostic detector, aligning with the methodology of prior research like CutLER, as mentioned in Line 210-212 of our paper. Consequently, when evaluating on LVIS, it is necessary for us to consider **objects from all categories**, irrespective of their categories as rare, common, or frequent.
>
> 7. Rigorous Definitions of Whole, Part, and Subpart
>
> We will revise our paper based on these concepts:
>
> - Whole: The entirety of an object, encompassing all its parts and subparts. In our method, a “whole” object is the largest coherent grouping of pixels that the algorithm identifies as an entity. For instance, when considering an image of a bicycle, the “whole” object would refer to the entire bicycle, including all its parts such as wheels, frame, and handle.
>
> - Part: A smaller component of the “whole” object that has consistent features but does not encompass the entire object. In the bicycle example, the wheel or the handle would be considered a “part” of the bicycle.
>
> - Subpart: An even smaller component of a “part”. It provides finer granularity in object segmentation. In the bicycle example, if we consider the wheel as a “part”, then the spokes and tire of the wheel could be considered “subparts”.
>
> 8. Overcoming Previous Limitations
>
> - Limitation 2 - Narrow object coverage: Many prior methods, including TokenCut and MaskCut from CutLER, can only detect a *pre-defined number of objects* in images due to their algorithm design. However, our hierarchical adaptive clustering (Section 3.1) can recognize a broader range of objects, with the *number adaptively based on image content*. This approach markedly improves object detection, surpassing prior methods. As shown in Table 1, we boost Average Recall (AR) for self-supervised methods on SA-1B from 17.0 to 26.0.
>
> - Limitation 3 - Inefficiency: Prior methods like FreeSOLO and CutLER involve *multiple rounds of self-training*, increasing computational demands. For the first time, we introduced the Mean Teacher paradigm to our framework (Section 3.3), which allows a teacher model *continually produce pseudo-labels*, with a student model *simultaneously learning*. Compared to traditional multi-round self-training, our approach is smooth and efficient. Our method takes only 1/12 of the training iterations (Line 205) to outperform CutLER.

---

> > ### Author Response · Authors · 2023-08-17
> > **Follow-up on LVIS Metrics**
> >
> > As clarified in our previous response, our main objective is to identify **all objects from all categories** present in scene images without distinguishing based on classes, so we did not evaluate LVIS-rare/common/frequent metrics, which are typically used in a different task of class-aware object detection.
> >
> > While we do believe that the full LVIS is more suitable as the evaluation benchmark for our class-agnostic detection task, per the reviewer’s request, we have evaluated our model with the LVIS-rare/common/frequent metrics, and compared it against the prior state-of-the-art method CutLER. The results are summarized in the following table. Again, we demonstrate improved performance in every metric, which is consistent with our previous results. In contrast to class-aware object detection, we do not observe a significant gap between the detection performance for rare/common/frequent objects, since our method is developed in a **self-supervised and class-agnostic** manner.
> >
> > |           	| AR$_r$  | AR$_c$  | AR$_f$  | AR   |
> > | ------------- | ---- | ---- | ---- | ---- |
> > | CutLER    	| 17.9 | 22.2 | 20.2 | 20.2 |
> > | HASSOD (Ours) | 20.8 | 23.9 | 22.6 | 22.5 |

---

> > > ### Author Response · Authors · 2023-08-19
> > >
> > > Hope this message finds you well. This is just a friendly reminder of our recent rebuttal and follow-up responses, in which we have addressed your valuable feedback on our work.
> > >
> > > Could you please take a moment to review our responses? If any issues remain unresolved or further clarification is needed, we are more than willing to continue the discussion.
> > >
> > > Thank you once again for your time and expertise. We look forward to hearing from you soon.

---

> ### Comment · Reviewer_wiMX · 2023-08-21
>
> Sorry for the delay. Thanks for the effort of the authors. Most of my concerns have been solved. I increased the score.

---

> > ### Author Response · Authors · 2023-08-21
> >
> > We are glad that our responses have addressed your concerns. We appreciate your helpful feedback and reconsideration of the rating.

---

### Official Review · Reviewer_cMyQ · 2023-06-30

**Soundness:** 3 good
**Presentation:** 3 good
**Contribution:** 2 fair
**Rating:** 5
**Confidence:** 4

**Summary:**

The authors propose a self-supervised object detection approach by employing a self-supervised pre-trained model (i.e., DINO) to hierarchically and adaptively group regions into object masks using multiple pre-defined thresholds based on cosine similarity in the feature space. The authors then adapt the Mean Teacher framework to train a student object detector with the initial pseudo-labels from the clustering process as well as the progressively refined pseudo-labels from the teacher model. Extensive experiments demonstrate the superiority of the proposed approach over existing self-supervised object detection and instance segmentation methods.

**Strengths:**

-	The experiments are extensive and the results are promising.
-	The paper is generally well-written and easy to follow.

**Weaknesses:**

-	The authors use a set of pre-defined thresholds (i.e., {0.1, 0.2, 0.4}) to merge region features. However, considering different datasets may have different distributions, such COCO-tuned hyper-parameters may not be suitable when merging region features for other datasets. Thus, the authors have to select the optimal thresholds for each dataset when performing clustering, which limits the practical applications of the proposed method to some extent.
-	The authors use some post-processing techniques like CRF to refine the masks. I am curious about the mask quality without such post-processing. Will the performance be dropped significantly?
-	Although the proposed method can segment an image with hierarchy, it seems that the mask quality is somewhat inferior to CutLER according to the qualitative results as shown in the paper.
-	The authors choose DINO with ViT-B/8 model, i.e., each spatial element in the feature map corresponds to a 8x8 patch in the original image. What is the effect of the patch size? It seems that the patch size tends to affect the performance significantly and should also be ablated.
-	The authors use Cascade Mask R-CNN to verify the effeteness of the proposed method. What about other kinds of detectors? It would be better to provide results based on more detectors to verify the generality of the proposed method.

**Questions:**

See questions mentioned above. I am currently leaning towards borderline reject and hope the authors could address my concerns during the rebuttal.

**Limitations:**

The authors have well addressed the limitations and the broader impacts in Sec. 5, which looks good to me.

---

> ### Author Rebuttal · Authors · 2023-08-10
>
> We appreciate the detailed feedback you provided for our submission. We are encouraged by your acknowledgement that our “experiments are extensive”, “results are promising”, and “paper is well-written and easy to follow”. We provide the following clarifications in response to your concerns:
>
> 1. Generalizability of Pre-Defined Thresholds
>
> Please check our general response.
>
> 2. Mask Quality Without CRF
>
> The use of the CRF post-processing is necessary for acquiring precise edges in pseudo-labeled masks. When generating initial pseudo-labels, we employ ViT-B/8 to extract features at the patch-level, with each patch being of the size 8x8. Absent CRF post-processing, the masks would inherently be composed of these 8x8 patches. As a consequence, the boundaries of such masks would resemble a jigsaw puzzle and would not be consistent with the real boundaries of objects. Meanwhile, we adopt CRF post-processing following the foundational segmentation work DeepLab [Ref1], as well as prior methods on self-supervised object detection including TokenCut and CutLER. For example, CRF post-processing has been shown necessary in Figure 4 of TokenCut [Ref2].
>
> 3. Inferior Mask Quality
>
> It is true that there are some instances where the quality of masks generated by our method may seem inferior when compared directly to CutLER. There are a couple of reasons for this:
>
> - Training duration: As indicated in Section 4.1, our model was trained for a considerably shorter duration compared to CutLER – specifically only **1/12** of CutLER's total iterations. This was mainly due to our computational resource constraints. We believe that with prolonged training, our model has the potential to produce higher quality masks.
>
> - Focus on overall performance: It is also crucial to highlight that our primary objective was to improve the overall detection and segmentation performance by increasing the number of detected objects, especially in challenging scenarios. While we also strive for mask perfection, our main focus is still on improving the overall average recall and precision. As evidenced in Table 1, our approach does demonstrate superior overall performance in instance segmentation, which we consider a significant achievement.
>
> 4. Impact of Patch Size in DINO ViT-B/8
>
> We appreciate your query regarding the ViT patch size and have produced the following table as additional ablation study. For fair comparison, we use 480x480 input resolution for ViT-B/8 and 960x960 for ViT-B/16 so that they have the same number of patches. With the same $\theta^\text{merge}$, ViT-B/16 leads to slightly fewer labels per image, but the quality is significantly worse than ViT-B/8, especially for small and medium objects. Therefore, we apply ViT-B/8 in our experiments for its localized visual features and subsequent high-quality pseudo-labels.
>
> | DINO Backbone | $\theta^\text{merge}$ | Labels per Image | AR | AR$_S$ | AR$_M$ | AR$_L$ | AP |
> |---|---|---|---|---|---|---|---|
> | ViT-B/8  | 0.1 | 2.58 | 4.1 | 0.6 | 5.1 | 15.6 | 1.8 |
> | ViT-B/16 | 0.1 | 1.97 | 3.0 | 0.3 | 2.3 | 14.6 | 1.1 |
> | ViT-B/8  | 0.2 | 4.20 | 5.3 | 0.9 | 7.0 | 18.4 | 1.8 |
> | ViT-B/16 | 0.2 | 3.19 | 3.8 | 0.4 | 3.3 | 17.4 | 1.2 |
> | ViT-B/8  | 0.4 | 11.61| 7.8 | 1.7 | 12.1| 22.1 | 1.3 |
> | ViT-B/16 | 0.4 | 10.15| 5.7 | 0.9 | 6.6 | 21.7 | 1.3 |
>
> 5. Use of Cascade Mask R-CNN
>
> - We chose Cascade Mask R-CNN to ensure **fair comparison with CutLER**. The same detector architecture enables a fair and direct comparison between our proposed method and CutLER. Otherwise, it would be unclear whether the performance gain came from the improved learning paradigm or a stronger detector architecture.
>
> - We do acknowledge the importance of demonstrating the versatility of our method across different detectors. To this end, we are conducting experiments using our proposed method and Mask R-CNN, and we will share the results within the next week.
>
> [Ref1] Chen et al. DeepLab: Semantic Image Segmentation with Deep Convolutional Nets, Atrous Convolution, and Fully Connected CRFs. In TPAMI, 2017.
> [Ref2] Wang et al. TokenCut: Self-supervised Transformers for Unsupervised Object Discovery using Normalized Cut. In CVPR, 2022.

---

> > ### Author Response · Authors · 2023-08-17
> > **Follow-up on Use of Cascade Mask R-CNN**
> >
> > As clarified in our previous response, we chose Cascade Mask R-CNN mainly to ensure **a fair comparison with CutLER**. Meanwhile, we do acknowledge the importance of demonstrating the versatility of our method across different detectors. To this end, we have trained a **Mask R-CNN** model using our proposed method. The following table compares LVIS detection performance of Mask R-CNN trained by CutLER and HASSOD. The results demonstrate that our method is indeed generalizable and consistently outperforms CutLER with different detector architectures. We will include the results in the revised version as well.
> >
> > |           	| Box AR | Box AR$_S$ | Box AR$_M$ | Box AR$_L$ | Box AP | Mask AR | Mask AR$_S$ | Mask AR$_M$ | Mask AR$_L$ | Mask AP |
> > | ------------- | ------ | ------- | ------- | ------- | ------ | ------- | -------- | -------- | -------- | ------- |
> > | CutLER    	| 20.7   | 10.4	| 33.3	| 52.0	| 4.1	| 18.5	| 9.6  	| 29.1 	| 44.9 	| 3.4 	|
> > | HASSOD (Ours) | 23.8   | 13.5	| 38.3	| 50.9	| 4.3	| 21.5	| 11.8 	| 35.1 	| 46.6 	| 4.1 	|
> >
> > In summary, while our primary focus was on Cascade Mask R-CNN for a consistent comparison with CutLER, our method is certainly not restricted to it and shows superior performance with other detectors as well.

---

> > > ### Author Response · Authors · 2023-08-19
> > >
> > > Hope this message finds you well. This is just a friendly reminder of our recent rebuttal and follow-up responses, in which we have addressed your valuable feedback on our work.
> > >
> > > Could you please take a moment to review our responses? If any issues remain unresolved or further clarification is needed, we are more than willing to continue the discussion.
> > >
> > > Thank you once again for your time and expertise. We look forward to hearing from you soon.

---

> > > > ### Comment · Reviewer_cMyQ · 2023-08-19
> > > > **Response to rebuttal**
> > > >
> > > > Thanks for the rebuttal. I have read the authors' response and the other reviewers' comments. Overall, I think this is a borderline paper given the current status of the paper. I would like to raise my score to borderline accept to reflect the efforts the authors made during the rebuttal. The paper could be made much stronger if the authors could revise their paper based on the discussion during the rebuttal.

---

> > > > > ### Author Response · Authors · 2023-08-19
> > > > >
> > > > > Thank you very much for your time and thoughtful consideration of our work. We deeply appreciate you raising the score based on our rebuttal. Following your constructive feedback, we are committed to further revising and strengthening the paper, by incorporating all of our discussion during the rebuttal.
> > > > >
> > > > > We would also like to highlight the growing interest within the object recognition community in class-aganotic object detection and instance segmentation, as exemplified by the emergence of models like the Segment Anything Model (SAM). In contrast to the supervised nature of SAM which extensively relies on human supervision, **our work shows the promise of a fully self-supervised direction** with strong, generalizable performance. As you have commented in the initial review, “extensive experiments demonstrate the superiority of the proposed approach over existing self-supervised object detection and instance segmentation methods.” We hope our self-supervised approach can benefit the research community timely and profoundly.

---

### Official Review · Reviewer_p4z2 · 2023-07-02

**Soundness:** 3 good
**Presentation:** 3 good
**Contribution:** 2 fair
**Rating:** 4
**Confidence:** 5

**Summary:**

This goal of this paper is to improve the performance of self-supervised object detection by enhancing the detection pseudolabels. In addition to semantic masks provided as detection labels to an object detector, the authors utilize a rule-based approach to automatically generate a hierarchical structure of objects and provide it as targets to the detector. By training on hierarchical pseudolabels, the detector learns about the compositional structure of objects and therefore performs better. Additionally, the authors use a teacher-student network to train the detector instead of multi-round self-training approach. The proposed approach outperforms two other self-supervised detection approaches.

**Strengths:**

- The authors tackle an important problem in perception; the hierarchical structure of objects. By providing object labels at multiple levels in a tree structure, the detector can learn patterns in compositional structures of objects (subparts - parts - whole), which helps improve the predictions. For example, a model can learn to condition a whole object prediction on the existence of its parts and subparts.
- This hierarchical structure is not only found in vision, but also other perceptual modalities and NLP. Therefore, understanding how to generate and utilize this structure can be helpful in other tasks and across other modalities.
- This method is completely self-supervised, based on features extracted from DINO trained on ImageNet. The improvement in detection performance comes at no supervision cost, only additional computational cost to generate the pseudolabels.
- The approach shows promising quantitative results, surpassing other self-supervised methods.

**Weaknesses:**

- This approach, like many others, will always be upper bounded by the quality of the features and the robustness of the self-supervised representation learning approach used to generate them. The generation of labels depends on frozen features, so the detector will be sensitive to the quality of those features.
- Approaches that use hierarchical clustering lose representation quality as they naively average high dimension feature representations to merge clusters. Therefore these methods cannot cluster the object representations to generate class labels for training. This is similar to approaches like TW-FINCH [1] that takes the same route for event segmentation. This being said, I'm interested in seeing whether high-level representations of objects generated by averaging low-level pixel/patch representation would provide good matching to true class labels. An alternative would be to add a (non-)linear on the average-based representations to classify the objects.
- While there is no supervision cost to generate the extra hierarchical labels, there is computational cost in clustering and structure discovery. It is important to share the computational cost in order to determine whether this approach is practical and can be scaled up.
- I am not convinced that simple thresholding of the similarity metric provides accurate separation between the hierarchy levels. Using an arbitrary threshold will guarantee a tree structure but not necessarily a part-whole structure. The goal is not to find any tree structure whith parent masks covering children masks. These rules do not guarantee a part-whole structure. Each entity in the tree should be a coherent object (part-whole tree).
- One suggestion is to plot $\theta^{merge}$ against the number of objects in the image and detect regions with lowest slopes to be the thresholds. The intuition is that in these regions the model is less sensitive to changes in $\theta$, therefore there could be good separation between hierarchical levels at these thresholds.
- How do the authors decide on $\theta^{merge}$? Is it using the ablations in Table 2? Table 2 is generated using groundtruth masks. So if the decision on $\theta^{merge}$ is based on evaluating the masks with ground truth masks, this would be a significant flaw in the experimental protocol. It would violate the self-supervision claim. Usually researchers decide based on a small validation set, and even then it is still arguably supervised. This is an important point.

[1] Sarfraz, Saquib, et al. "Temporally-weighted hierarchical clustering for unsupervised action segmentation." Proceedings of the IEEE/CVF Conference on Computer Vision and Pattern Recognition. 2021.

**Questions:**

- When using a single threshold to generate the masks, my understanding is that we should only get one level of objects. In other words, the output masks should not intersect with any other masks. How are the numbers in Table 2 generated when using a single $\theta^{merge}$ threshold? This approach should only be possible when having intersecting masks (i.e., ensemble).

**Limitations:**

- Limitations about not aligning hierarchical levels with human perception are addressed. But it's missing limitations concerning the inability of this model to generate representations useful to classification of the detected boxes/masks.

---

> ### Author Rebuttal · Authors · 2023-08-10
>
> We appreciate the detailed feedback you provided for our submission. We are encouraged by your acknowledgement that our method tackles “an important problem of hierarchical structure of objects”, is “completely self-supervised”, and shows “promising quantitative results”. We provide the following clarifications in response to your concerns:
>
> 1. Upper Bound of Self-supervised Representation
>
> HASSOD is **not upper bounded** by such representations.
>
> - Starting from existing self-supervised representations is a standard practice in the literature of self-supervised object detection. For instance, FreeSOLO utilizes pre-trained DenseCL representations, and the state-of-the-art method CutLER relies on DINO representations. Pre-trained self-supervised representations have been shown as a good starting point for self-supervised object detection. We are also following this paradigm.
>
> - The performance of our method is **not upper bounded** by the constraints of the frozen self-supervised models. Two key insights in HASSOD help us break through the limits of DINO features by self-improvement: 1) We train a detection model to learn common objects and their hierarchical relations for *enhanced generalization* to unseen images. 2) We leverage Mean Teacher for continual self-enhancement, and gradually *minimize the negative impact of noisy initial pseudo-labels*.
>
> - A close examination of Table 2 and Table 3 in our paper demonstrates that HASSOD is not upper bounded by DINO representations. Specifically, the quality of pseudo-labels generated by frozen DINO features can be quantified by the best average recall (AR) and average precision (AP) listed in Table 2 as 8.9 and 1.8, respectively. After training a detection model with HASSOD, these values are boosted to 22.4 and 6.3, respectively, as shown in Table 3.
>
> 2. Object Representations for Classification
>
> - Firstly, our primary task is **class-agnostic** self-supervised object detection, established by prior work such as FreeSOLO and CutLER. Distinct from the traditional class-aware object detection tasks, this direction is motivated by numerous scenarios where the primary goal is not to recognize specific object classes but rather to capture objects within an image. As examples in robotic vision, a robot may need to identify and navigate around obstacles without classifying them.
>
> - Meanwhile, we acknowledge the value in examining our extracted features for class-aware object detection. To this end, we tune a set of classification heads newly attached to the pre-trained Cascade Mask R-CNN model. The other modules are initialized from CutLER or HASSOD and frozen during the supervised fine-tuning on MS-COCO. The results of this classification-only tuning are shown in the table below. The model pre-trained by HASSOD achieves $2\times$ AR and AP in class-aware object detection and instance segmentation, suggesting that HASSOD has gained representations more helpful for classification during the self-supervised learning procedure, as compared with CutLER.
>
> |Initialization|AR|AR$_S$|AR$_M$|AR$_L$|AP|
> |---|---|---|---|---|---|
> |CutLER|4.2|1.1|2.8|6.3|1.5|
> |HASSOD (Ours)|**8.4**|**1.6**|**6.8**|**14.2**|**3.1**|
>
> 3. Computational Costs
>
> Please check our general response.
>
> 4. Simple Thresholding for Accurate Hierarchy Levels
>
> Your concerns over the use of thresholding for deriving hierarchical structures are valid. Indeed, using simple thresholding could potentially lead to deviations from the expected part-whole structure that we typically associate with objects. For instance, there might be instances where we generate a structure that is closer to an incomplete whole object rather than a distinct part.
>
> Despite this concern, thresholding in conjunction with the DINO features leads to surprisingly coherent and semantically meaningful results. Here are our observations:
>
> - Coherence of pseudo-labels: From our extensive experiments and observations, the pseudo-labels generated are **predominantly coherent**, displaying uniformity in color and edge characteristics. We rarely find significant heterogeneity in the pseudo-labels. In Figure 2, we showed a real example from our pseudo-labels. This essentially implies that the hierarchical structures produced by thresholding and DINO are *surprisingly consistent* with the part-whole structures that we, as humans, commonly perceive.
>
> - Robustness of DINO Features: The quality and efficiency of the hierarchical structures are highly attributed to the DINO features. Prior work [Ref1, 2, 3] has demonstrated DINO's ability in identifying **whole objects**. Impressively, DINO features are robust with respect to various types of thresholds. Our contribution is to extend this understanding further by not only segmenting whole objects but also extracting **meaningful and coherent parts of objects**.
>
> 5. Deciding and Plotting Thresholds $\theta^\text{merge}$
>
> Please check our general response.
>
> 6. Table 2 with Single $\theta^\text{merge}$ Threshold
>
> In Table 2, for the rows with one single $\theta^\text{merge}$ threshold, as the reviewer correctly pointed out, we would produce a collection of pseudo-labels, in which each mask does not intersect with others. However, Table 2 aims to evaluate the quality of these generated *initial pseudo-labels* as individual entities. The evaluation does not involve the hierarchical relationships between the labels. Instead, it provides an assessment of how well these pseudo-labels align with true object masks in the image based on the selected threshold. The hierarchical structure of masks is learned and assessed in the subsequent stage of HASSOD.
>
> [Ref1] Siméoni et al. Unsupervised Object Localization: Observing the Background to Discover Objects. In CVPR, 2023.
> [Ref2] Wang et al. Cut and Learn for Unsupervised Object Detection and Instance Segmentation. In CVPR, 2023.
> [Ref3] Hamilton et al. Unsupervised Semantic Segmentation by Distilling Feature Correspondences. In ICLR, 2022.

---

> > ### Comment · Reviewer_p4z2 · 2023-08-15
> > **Comment on the Rebuttal**
> >
> > Thank you for the rebuttal. I have read your answers to my questions and the other reviews.
> >
> > I am still not convinced that the performance is not upper-bounded by the **quality** of the frozen features you use for hierarchy discovery. My understanding is that the better the frozen features, the better the separation between hierarchical levels (as you mentioned in answer 4 about DINO features coherence). A better hierarchy improves object detection performance, because this is the premise of the paper. Therefore the improvement in object detection performance is always bounded by the quality of frozen features.
> >
> > As I mentioned in the review, I understand this procedure is similar to other methods that use pretrained frozen features and will lead to object detection not improving beyond a certain level, because the features are frozen. Showing that object detection performance trained on the hierarchy results in higher performance than objects detected from frozen features seems irrelevant to the concern I raised. If you cannot enhance the frozen features quality, you cannot improve the hierarchy quality, therefore cannot improve the object detection performance. Is my understanding correct, or am I missing something?
> >
> > Thank you!

---

> > > ### Author Response · Authors · 2023-08-17
> > > **Follow-up Response**
> > >
> > > Thank you for your insightful comment. We would like to further clarify our approach in light of your remaining concern about the pretrained features in the following aspects:
> > >
> > > 1. Focus on Object Detection, Not Representation Learning
> > >
> > > 	We would like to emphasize that the core of our research is on self-supervised *object detection*, rather than on improving self-supervised *representations* themselves. As the reviewer mentioned, consistent with previous state-of-the-art work including CutLER, our approach leverages existing pre-trained features to perform the object detection task. We have demonstrated that our approach HASSOD, while using the same pre-trained features by DINO, leads to enhanced results compared to prior works (see Table 1), highlighting the effectiveness of our specific contributions to self-supervised object detection.
> > >
> > > 2. Two Avenues for Performance Improvement
> > >
> > > 	Enhancing the frozen feature quality is *not the only way* to improve the object detection performance. In fact, from the perspective of pre-trained features, we believe that there are two ways to improve self-supervised object detection performance:
> > > 	- Approach 1: As you have correctly suggested, and we fully agree, one possible way for performance enhancement is through *better pre-trained features*. Currently, DINO has proven to be the *most effective pre-trained features for object discovery* and has been utilized by state-of-the-art methods such as CutLER. Future advances in self-supervised representation learning could yield even better pre-trained features, and using such improved features would further improve detection performance. However, developing more advanced self-supervised representations is beyond the scope of this paper.
> > > 	- Approach 2: Contrary to relying solely on frozen features, our work showcases the ability to **refine these features through the object detection training process**. We find this a critical contribution to highlight. In both HASSOD and existing self-supervised object detection work, the initial pseudo-labels generated by the frozen features could be noisy and coarse. For example, when the boundary between a foreground object and its background is vague, the frozen features are less distinguishable between the foreground and background, thus leading to inaccurate mask edges or even missed objects. However, after observing more similar instances in the whole dataset, the detector can gradually learn to fix errors in pseudo-labels, and detect such objects accurately. During this procedure, the pre-trained features are also adapted in an end-to-end manner for detecting more challenging objects. Therefore, we can achieve higher performance by training a detector than directly deriving objects from frozen features – this result was pointed out in our previous response. We hope our explanation here also clarifies the reviewer’s question regarding how this result is relevant to addressing the reviewer’s concern.
> > >
> > > 3. Refinement Beyond Frozen Features
> > >
> > > 	It is important to clarify that while we start our process with pre-trained DINO features, these features are **not permanently frozen** in our approach. During the training of our object detection model, we do fine-tune the DINO backbone. This fine-tuning, coupled with self-correction in detector training and the Mean Teacher strategy, enables us to adapt and potentially improve the DINO representations specifically for the object detection task.
> > >
> > > 	We provide a table below to compare the quality of objects discovered directly using the *frozen DINO ResNet-50 backbone* versus those discovered using the *fine-tuned backbone in our trained detector*. We use our hierarchical adaptive clustering strategy and standard post-processing to discover objects. Please note that this table is a direct comparison on the feature quality between the two backbones, rather than the final detection results. This comparison explicitly illustrates our ability to *substantially enhance* the pre-trained representations for the task of object discovery.
> > >
> > > 	| Backbone                               	| AR   | AR$_S$  | AR$_M$  | AR$_L$  |
> > > 	| ------------------------------------------ | ---- | ---- | ---- | ---- |
> > > 	| Frozen DINO ResNet-50                  	| 1.8  | 2.1  | 1.3  | 2.0  |
> > > 	| Fine-tuned DINO ResNet-50 in HASSOD (Ours) | 3.4  | 3.9  | 3.0  | 2.2  |
> > >
> > > In summary, our work not only leverages pre-trained representations, but also refines and adapts them to the specific task of self-supervised object detection.

---

> > > > ### Comment · Reviewer_p4z2 · 2023-08-18
> > > > **Follow up question**
> > > >
> > > > Thank you for the reply. I have a follow up question.
> > > >
> > > > Are the enhanced features by object detection used to generate a better hierarchy or is the hierarchy fixed based on the frozen DINO features?
> > > >
> > > > I understand that the method optimizes the DINO features during object detection training, but that happens during training of any task. If the enhanced features from object detection are not used to enhance the the hierarchy (perhaps in an iterative or a joint training algorithm), then your proposed idea of using a hierarchy to improve the object detection performance is still bounded by the quality of those features you used to create the initial fixed hierarchy, because this hierarchy is fixed and is not affected by any enhancement of the frozen features. Please clarify this point.
> > > >
> > > > Thank you.

---

> > > > > ### Author Response · Authors · 2023-08-18
> > > > >
> > > > > Yes, the enhanced features are indeed used in improving the hierarchical pseudo-labels. The hierarchical pseudo-labels are not fixed during our training.
> > > > >
> > > > > In our work, the enhanced DINO features engage in **providing better pseudo-labels through the process of Mean Teacher**. As introduced in Section 3.3, the student detector is supervised first by (a) fixed initial pseudo-labels, and the learning target gradually moves to (b) improved pseudo-labels produced by the teacher detector. The teacher detector contains the enhanced DINO backbone and utilizes its fine-tuned features. Please note that we do not have to actually rerun the hierarchical adaptive clustering algorithm on the enhanced DINO to discover hierarchical objects. Instead, since we have tuned a detector based on the enhanced DINO, **we can directly read out the predictions from the detector as the refined hierarchical pseudo-labels**.
> > > > >
> > > > > Generating improved pseudo-labels directly from the **trained teacher detector** is preferred, because it is more efficient than clustering DINO backbone features, and leads to hierarchical pseudo-labels of higher quality. A similar design has been applied in prior work such as FreeSOLO and CutLER (but with a multi-round iterative self-training routine that is inferior to our Mean Teacher). However, we cannot do so in the beginning when only the pre-trained DINO is available. Our detector model receives supervision from the combination of the (a) fixed initial pseudo-labels and (b) hierarchical pseudo-labels of higher quality from the teacher detector. Our adaptive target strategy modulates the weights for (a) and (b) during the training process, so that the impact of noises in (a) can be gradually reduced, the benefit of consistently improved (b) can be gradually magnified, and in the end, we are **not bounded by the quality of (a)**.

---

> > > > > > ### Comment · Reviewer_p4z2 · 2023-08-18
> > > > > >
> > > > > > Thank you for clarifying. I now better understand the proposed online process of refining the hierarchy.

---

> > > > > > > ### Author Response · Authors · 2023-08-19
> > > > > > >
> > > > > > > We sincerely appreciate the constructive discussion we’ve had, and we are glad our clarification has helped you understand our online refinement process.
> > > > > > > Considering the progress of our discussion and your latest positive comment, we kindly request you to reconsider the rating of our submission, if possible. Thank you once again for your valuable feedback.

---

> > > ### Comment · Reviewer_s64Y · 2023-08-18
> > > **Agree**
> > >
> > > I agree entirely agree with this view.
> > >
> > > Frozen Features have been explored a lot,  but analyzed deeply.

---

### Official Review · Reviewer_s64Y · 2023-07-07

**Soundness:** 3 good
**Presentation:** 3 good
**Contribution:** 3 good
**Rating:** 5
**Confidence:** 4

**Summary:**

This paper proposes a hierarchical Adaptive Self-Supervised Object Detection (HASSOD), an approach that  learns to detect objects and understand their compositions without human supervision.
HASSOD employs a hierarchical adaptive clustering strategy to group regions into object masks based on self-supervised visual representations, adaptively determining the number of objects per image.
HASSOD identifies the hierarchical levels of objects in terms of compositionality, by analyzing coverage
relations between masks and constructing tree structures.
The proposed method adapts the Mean Teacher framework from semi-supervised learning, which leads to a smoother and more efficient training process.

The Mask AR is improved from 20.2 to 22.5 on LVIS, and from 17.0 to 26.0 on SA-1B.



**Strengths:**

Good results of the process, exhibited  in Fig. 3.

Many quantitative comparisons in Tables 1-3.

Nice to read the overall paper.

Good to see the reference of the vital work of [7], even after more than a decade.

Appendix Doc highlights the importance of a better metric than AR.


**Weaknesses:**


Lot of experiments, results, but no analytics. Not even a single equation, cost function, hyper-parameter set mentioned.

Adaptation of mean-teacher model, still makes it a self-supervised framework.,

The overall designed architecture is explained with examples of images, in a diagram in fig. 2.
A flowchart would still be better.
Which part requires CNN-based learning, which part requires shallow learning (if any) or heuristics is
not so clear (except a bit in Sec. 4.1, a small 2nd para).

Only in the Appendix -
The platform/environment/resources used for implementation are specified.
The computational time required for each sub-stage of the process is also not mentioned,
including the inference time. This is very important for practitioners.

Few failure cases should have been highlighted with causes/reasons; Appendix is not thorough on this too.

Color labels used for inference vs bounding box, often causes confusion - the color overlap for anyone to clearly understand.



**Questions:**

If this proposed work is inline with self-supervision,
what are the equivalent/compatible stages of  -
pretext learning using base data with no labels;
fine-tuning with support set;
target set for inference on downstream task?

A table specifying the dataset distributions (although names given) and stages exploiting the same
for self-supervision should have been given.

Any scope of few-shot learning or meta-learning framework in your proposed approach?


**Limitations:**

Lack of analytics ;
NO ablation studies with cost functions;
Lack of mapping to standard Self-supervision paradigm, is a concern.

Object Detection in complex cases of overlap of objects, background clutter not provided.

I think its high time that researchers also look at identification/detection in camouflage (a personal opinion).

---

> ### Author Rebuttal · Authors · 2023-08-09
>
> We appreciate the detailed feedback you provided for our submission. We are encouraged by your acknowledgement of our “good results”, “quantitative comparisons”, and overall writing. We provide the following clarifications in response to your concerns:
>
> 1. Lack of Analytics and Ablation Study Regarding Cost Functions and Hyper-parameters
>
> - Our central contribution is a **new learning paradigm**, rather than innovating on cost functions. We adopted a *minimalistic* design, using standard cost functions rooted in established literature. Most loss functions and hyper-parameters are inherited from previous works.
>
> -  To demonstrate the standard practices in our design, we can look into the two-stage learning of HASSOD (see more details in the general response):
>     - Stage 1: We apply the hierarchical adaptive clustering strategy, leveraging the DINO representations to derive initial pseudo-labels. The similarity measure in this process, as described in Line 118, involves computing the *pairwise cosine similarity* between adjacent region features: $\frac{x^T y}{\|\|x\|\|_2\|\|y\|\|_2}$. This choice of metric is well-grounded in the self-supervised learning literature such as SimCLR and MoCo.
>    - Stage 2: This stage focuses on training our object detection and instance segmentation model Cascade Mask R-CNN. We optimize *conventional detection and segmentation losses*, including 1) the foreground/background classification loss, 2) bounding-box regression loss, and 3) mask prediction loss, following Mask R-CNN and Cascade R-CNN.
>
> - Presentation choices: In our initial submission, we consciously chose to minimize the use of equations, especially when the textual description was sufficient. Our goal was to maintain the paper's readability and prevent readers from being overwhelmed by equations. However, in light of your feedback, we will revise our paper with a more explicit introduction to the computational procedures, backed by relevant equations and analytics.
>
> - Ablation study clarification: Since most of our chosen loss functions and hyper-parameters are rooted in standard practice, we found it unnecessary to ablate these specific choices. Meanwhile, we had ablation studies for **our novel designs**: hierarchical level prediction (Section 3.2) and mean teacher training with an adaptive target (Section 3.3). The details are in Table 3.
>
> 2. Self-Supervised Nature with Mean Teacher
>
> We designed our method to be fully self-supervised relying on no labeled data, which we perceive as *a primary advantage*. We would be grateful for additional insights if there are specific concerns about our self-supervised strategy.
>
> 3. Figure 2 and Clarification on Components Involving CNN Learning
>
> We have revised this figure and included it as Figure R1 in the rebuttal PDF. As mentioned in the general response, HASSOD is a two-stage *discover-and-learn* approach. CNN-based learning happens in the second stage.
>
> 4. Computational Details
>
> Please check our general response.
>
> 5. Lack of Failure Case Analysis
>
> We appreciate your suggestion and have included some failure case analysis in Figure R4 in the rebuttal PDF.
>
> 6. Overlap of Colored Masks and Bounding Boxes During Visualization
>
> We followed standard visualization practices for instance segmentation as Mask R-CNN. The perceived clutter in our figures stemmed from depicting challenging cases with multiple objects in a scene. Our hierarchical predictions also resulted in overlapped masks. Based on your suggestion, we will omit colored bounding boxes and only show segmentation masks for better visual clarity. As an example, please check the failure case visualization mentioned above.
>
> 7. Alignment with Standard Stages of Self-Supervision Paradigm
>
> - First, it is essential to note that our work does not delve into self-supervised **representation** learning, which is merely a subset of the broader self-supervised learning domain. Our primary focus is on self-supervised **object detection**, an inherently significant task in its own right. This task has a wide range of potential applications such as robotic vision. When contextualizing our work within the existing literature, our method aligns with the *discover-and-learn* paradigm by prior work such as FreeSOLO and CutLER.
>
> - The goal of our self-supervised learning task sets it apart from representation learning methods like SimCLR or MoCo. However, for the sake of the best understanding, the training of our model using the pseudo-labels produced by our clustering can be considered as the “pre-text” task, though the task goals are not generated on-the-fly. Moreover, our work does not involve any “fine-tuning” or “linear probing” on ground-truth labels, as our model intrinsically produces object-level bounding boxes and masks. Regarding “downstream evaluation”, we assess our model's performance on unseen datasets, like SA-1B, in a zero-shot setting.
>
> 8. Dataset Specifics for Self-Supervision
>
> Throughout our self-supervised learning process, we solely used the `train` and `unlabeled` splits of MS-COCO, which contains 0.24 million images. This was detailed in Section 4.1.
>
> 9. Incorporation of Few-Shot Learning or Meta-Learning
>
> While few-shot and meta-learning are valuable, our current research centers on fully self-supervised object detection. Integrating these techniques would be an interesting future exploration.
>
> 10. Object Detection in Complex Scenarios
>
> Indeed, our model has been tested in challenging cases with object overlap and cluttered backgrounds. Please refer to our supplementary material, particularly Figure 6 (first row) and Figure 7 (last two rows). These illustrations demonstrate the advantage of our model over CutLER in complex scenarios.

---

> > ### Author Response · Authors · 2023-08-10
> > **Follow-up on Detection in Camouflage**
> >
> > Thank you for highlighting Camouflage Object Detection (COD) [Ref 1]. COD presents a unique set of challenges, given the intrinsic similarities between the target and its environment. Currently, the best approaches to COD [Ref 1, 2] are based on high-quality human annotations and supervised learning. Given this context, utilizing self-supervised learning may not be immediately suitable. Exploring the overlap of self-supervised learning and COD is indeed an interesting future avenue.
> >
> > [Ref1] Deng-Ping Fan, Ge-Peng Ji, Guolei Sun, Ming-Ming Cheng, Jianbing Shen, Ling Shao. Camouflaged Object Detection. In CVPR, 2020.
> > [Ref2] Deng-Ping Fan, Ge-Peng Ji, Ming-Ming Cheng, Ling Shao. Concealed Object Detection. In TPAMI, 2022.

---

> > > ### Author Response · Authors · 2023-08-19
> > >
> > > Hope this message finds you well. This is just a friendly reminder of our recent rebuttal and follow-up responses, in which we have addressed your valuable feedback on our work.
> > >
> > > Could you please take a moment to review our responses? If any issues remain unresolved or further clarification is needed, we are more than willing to continue the discussion.
> > >
> > > Thank you once again for your time and expertise. We look forward to hearing from you soon.

---

### Author Rebuttal · Authors · 2023-08-10

In this response, we provide clarification on the common questions and concerns raised by the reviewers.

1. Clarification on Learning Procedure (Reviewers s64Y, wiMX)

Overall, HASSOD adopts a two-stage *discover-and-learn* approach, as illustrated in Figure R2 in the rebuttal PDF. This two-stage approach is not only intuitive but also a standard practice in the self-supervised object detection domain, as evidenced by literature such as FreeSOLO and CutLER.

- Stage 1 - Initial pseudo-label discovery: We employ our hierarchical adaptive clustering strategy (Section 3.1) to derive initial pseudo-labels. This process is based on a *frozen* DINO model, and it does *not* require learning any parameters.

- Stage 2 - Object detector learning: We *train a Cascade Mask R-CNN model* using the pseudo-labels from the first stage, enhanced by our hierarchical level prediction (Section 3.2) and Mean Teacher self-training (Section 3.3). This model has learned to detect and segment objects, offering *enhanced generalization* to images it has not seen during training, since it can learn from common objects and their relations in different training images.

2. Computational Costs (Reviewers s64Y, p4z2)

We apologize for any oversight in clearly presenting the computational platform, resources, and processing times within the main body of the paper. While we did mention some details regarding the *training data and iterations* in Section 4.1 and provided an extended discussion on the *computational platform* in Section F of the supplementary material, we understand the necessity for more comprehensive information. To address this, we have prepared the following table regarding the computation costs in pseudo-label generation.

|Step|Time Cost (sec/image)|Workers|Parallelized Cost (sec/image)|
|---|---|---|---|
|Merge and Post-process|11.7|8|1.46|
|Ensemble and Split|2.1|16|0.13|
|Total|13.8|-|1.59|

We use both the `train` and `unlabeled` splits of MS-COCO, totaling to about 0.24 million images. On computation nodes equipped with $4\times$ NVIDIA A100 GPUs, we can parallelize the processing of images, reducing the time to 1.59 sec/image. With 4 such nodes, we can complete the pseudo-labels generation for 0.24 million images in $1.59\times0.24\times10^6/86400/4\approx1$ day. Our method can be readily extended to an even larger scale of data with more parallel compute nodes.

For model training, the Cascade Mask R-CNN training procedure takes about 20 hours on our node with $4\times$ NVIDIA A100 GPUs. During inference, each image takes 0.15 second on average.

We will include the computational information in our revision.

3. Deciding Thresholds $\theta^\text{merge}$ (Reviewers p4z2, cMyQ)

Instead of relying on the validation performance shown in Table 2, we chose the merging thresholds $\theta_i^\text{merge}$ purely based on computational considerations and empirical observations, ensuring that our approach is fully self-supervised.

- Guidance by number of pseudo-masks: Our choice for $\theta^\text{merge}$ was primarily guided by the **number of pseudo-label masks produced per image**. In the rebuttal PDF, Figure R3 shows the relationship between #masks/image and different thresholds. When $\theta^\text{merge} \ge 0.5$, the number of masks per image escalates rapidly. This steep increase incurs *significant computational costs*, both during the initial generation of pseudo-labels and the subsequent data loading and pre-processing procedures during model training. To strike a balance between computational efficiency and the desired mask granularity, thresholds of $\theta_i^\text{merge} \in \{0.1, 0.2, 0.4\}$ were chosen. This choice is consistent with the suggestion by Reviewer p4z2.

- Threshold generalization across datasets: Another noteworthy observation is the generalizability of these thresholds across various datasets. In the following table, we present the **number of generated pseudo-labels per image** on three datasets. With the merging threshold $\theta^\text{merge}$ fixed, the number of generated labels is relatively stable, regardless of the source image dataset. Therefore, our pre-set thresholds are generalizable and require no further tuning when transferred to other datasets. Meanwhile, our detection model was trained on MS-COCO images with pseudo-labels generated using our $\theta_i^\text{merge}$, and could generalize well to other datasets in a zero-shot manner, as shown in Table 1. This fact shows that the $\theta_i^\text{merge}$ are effective regardless of evaluation datasets.

|$\theta^\text{merge}$|MS-COCO|Objects365|SA-1B|
|---|---|---|---|
|0.1|2.58|3.49|2.91|
|0.2|4.20|5.78|4.88|
|0.4|11.61|12.15|12.70|

---

### Decision · Program_Chairs · 2023-09-21

**Decision:**

Accept (poster)

**Comment:**

This paper addresses the problem of class-agnostic object detection under the self-supervised learning setting by leveraging DINO's capability of forming the hierarchical whole-part pseudo labels to train cascade Mask R-CNN using the mean-teacher pipeline. The paper has been reviewed by several knowledgeable reviewers. The majority consensus from the reviews was that the paper will be of interest to the community and should be accepted. The reviewers acknowledged the authors' rebuttal efforts very positively. This meta-review concurs, and recommends acceptance.

However, as noted by the reviewers, there are some issues that need to be addressed carefully in revision, e.g.,

Whether the performance of the proposed method will "be upper bounded by the quality of the features and the robustness of the self-supervised representation learning approach used to generate them" (p4z2). The authors and p4z2 had in-depth discussions, which should be carefully justified and integrated into the main paper in revision.

Lack of analytics ; NO ablation studies with cost functions (s64Y)

Clarification on learning procedure (s64Y and wiMX). The revised figures in the rebuttal should be presented in the main paper in revision with precise explanations.